# Dis3L2 regulates cell proliferation and tissue growth through a conserved mechanism

**Benjamin P. Towler** [1]*, **Amy L. Pashler**[1], **Hope J. Haime** [1], **Katarzyna M. Przybyl**[1], **Sandra C. Viegas** [2], **Rute G. Matos**[2], **Simon J. Morley**[1], **Cecilia M. Arraiano**[2], **Sarah F. Newbury** [1]*

1 University of Sussex, Falmer, Brighton, United Kingdom, 2 Instituto de Tecnologia Química e Biológica António Xavier, Universidade Nova de Lisboa, Oeiras, Portugal

* b.towler2@bsms.ac.uk (BPT); s.newbury@bsms.ac.uk (SFN)

**Data Availability Statement:** Raw RNA-sequencing files have been deposited in ArrayExpress (https://www.ebi.ac.uk/arrayexpress/ ). Accession number: E-MTAB-7451.

## Abstract

Dis3L2 is a highly conserved 3'-5' exoribonuclease which is mutated in the human overgrowth disorders Perlman syndrome and Wilms' tumour of the kidney. Using *Drosophila melanogaster* as a model system, we have generated a new *dis3L2* null mutant together with wild-type and nuclease-dead genetic lines in *Drosophila* to demonstrate that the catalytic activity of Dis3L2 is required to control cell proliferation. To understand the cellular pathways regulated by Dis3L2 to control proliferation, we used RNA-seq on *dis3L2* mutant wing discs to show that the imaginal disc growth factor Idgf2 is responsible for driving the wing overgrowth. IDGFs are conserved proteins homologous to human chitinase-like proteins such as CHI3L1/YKL-40 which are implicated in tissue regeneration as well as cancers including colon cancer and non-small cell lung cancer. We also demonstrate that loss of DIS3L2 in human kidney HEK-293T cells results in cell proliferation, illustrating the conservation of this important cell proliferation pathway. Using these human cells, we show that loss of DIS3L2 results in an increase in the PI3-Kinase/AKT signalling pathway, which we subsequently show to contribute towards the proliferation phenotype in *Drosophila*. Our work therefore provides the first mechanistic explanation for DIS3L2-induced overgrowth in humans and flies and identifies an ancient proliferation pathway controlled by Dis3L2 to regulate cell proliferation and tissue growth.

## Author summary

Regulation of cell proliferation is not only important during development but also required for repair of damaged tissues and during wound healing. Using human kidney cells as well as the fruit fly *Drosophila* we have recently discovered that cell proliferation can be regulated by a protein named Dis3L2. Depletion or removal of this protein results in excess proliferation. These results are relevant to human disease as DIS3L2 has been shown to be mutated in an overgrowth syndrome (Perlman syndrome) where affected children have abnormal enlargement of organs (e.g. kidneys) and susceptibility to Wilms' tumour (a kidney cancer). Dis3L2 is an enzyme known to "chew up" mRNA molecules which instruct the cell to make particular proteins. Using state-of-the-art molecular

**Funding:** This work was funded by a Brighton and Sussex Medical School studentship [WC003-11 to B.P.T] and a University of Brighton studentship [WC003-30] to A.L.P. The work was also supported by a Genetics Society Summer Studentship [G2162] to H.H (https://genetics.org.uk/). and a Biochemical Society Summer Vacation Studentship [G2431] to K.M.P (https://biochemistry.org/). B.P.T was supported by a Biotechnology and Biological Sciences Research Council grant (BB/P021042/1) to S.F.N (https://bbsrc.ukri.org/). Work at ITQB NOVA was financially supported by: Project LISBOA-01-0145-FEDER-007660 (Microbiologia Molecular, Estrutural e Celular) funded by FEDER funds through COMPETE2020 (COMPETE 2020) - Programa Operacional Competitividade e Internacionalização (POCI) and by national funds through FCT - Fundação para a Ciência e a Tecnologia (https://www.fct.pt/); project PTDC/BIA-MIC/1399/2014 to C.M.A, project PTDC/BIM-MEC/3749/2014 to S.C.V and project PTDC/BIA-BQM/28479/2017 to R.G.M. R.G.M was financed by an FCT contract (ref. CEECIND/02065/2017); S.C.V was financed by program IF of "Fundação para a Ciência e a Tecnologia" (ref. IF/00217/2015). The Funders had no role in the study design, data collection and analysis, decision to publish or preparation of the manuscript.

**Competing interests:** The authors have declared that no competing interests exist.

methods in *Drosophila*, we have discovered that Dis3L2 targets a small subset of mRNAs, including an mRNA encoding a growth factor named 'imaginal disc growth factor 2' (*idgf2*). For human kidney cells in culture, we have found that depletion of DIS3L2 results in enhanced proliferation, and that this involves a well-known cellular pathway. Our results mean that we have discovered a new way of controlling cell proliferation, which could, in the future, be used in human therapies.

## Introduction

The coordinated control of cell proliferation is crucial during tissue growth in order to specify the size and shape of the adult organism. Control of cell proliferation is also required during tissue regeneration and wound healing. This regulation is important as lack of communication between cells to control proliferation may lead to tissue overgrowth or cancer. We have previously shown that the highly conserved 3'-5' exoribonuclease Dis3L2 is a new player in the co-ordinated control of cell proliferation [1]. Dis3L2 is a cytoplasmic member of the RNase II/RNB family of 3'-5' exoribonucleases which is conserved from bacteria through to humans [2,3]. Unlike the other eukaryotic members of this family (Dis3 and Dis3L1) Dis3L2 lacks the N-terminal PIN domain [2,3] which confers endoribonucleolytic activity and is required for interaction with the multicomponent exosome complex [4,5]. Dis3L2 also differs from Dis3 in that it shows high affinity for transcripts containing non-template Uridine at their 3' ends, a process catalysed by Terminyl Uridylyl Transferases (TUTases) [3,6–8]. Loss of function mutations in *DIS3L2* have been shown to result in the congenital overgrowth condition Perlman syndrome [9] in addition to a kidney tumour named Wilms' tumour. Whilst Perlman syndrome patients show a high predisposition to Wilms' tumour, up to 30% of sporadic Wilms' tumours were also shown to contain partial or complete deletion of *DIS3L2* [9]. In addition, deletions in *DIS3L2* have also been associated with skeletal overgrowth [10]. Therefore, understanding the *in-vivo* molecular mechanisms behind this will both aid our understanding of normal cell proliferation and these overgrowth diseases.

A number of experiments, using human, mouse, fly and *S. pombe* cells have suggested that the majority of natural Dis3L2 targets are non-coding RNAs such as tRNAs, snRNAs and snoRNAs [1,2,6,8,11–14]. In human cell lines and mouse embryonic stem cells these include the pre-miRNA *pre-let-7* as well as other non-coding RNAs (e.g. *Rmrp*) which have been shown to be polyuridylated and then degraded by Dis3L2 [6,12]. Recent work in human cells has demonstrated a role for Poly(A)-specific Ribonuclease (PARN) in protecting miRNAs from DIS3L2 mediated decay thus implicating mature miRNAs as other DIS3L2 targets [15]. Other direct targets which immunoprecipitated with *Drosophila* Dis3L2 included 5S rRNA and extended versions of *RNase MRP:RNA* [13]. This study, along with others, has demonstrated several roles for Dis3L2 in the removal of transcripts with incorrect processing, faulty transcriptional termination or those undergoing non-sense mediated decay [12,13,16]. However, it is currently unclear how defective degradation of the identified transcripts can enhance cell proliferation. A Dis3L2 knockout mouse model has demonstrated a transcriptional increase in *insulin growth factor 2* (*Igf2*) mRNA, however no overgrowth was observed [17], which may be due to perinatal mortality. Therefore, the cellular pathways linking Dis3L2 with proliferation have remained obscure.

Our previous work, using the model organism *Drosophila*, demonstrated that knockdown of Dis3L2 in the wing imaginal disc results in substantial wing overgrowth due to increased cellular proliferation during the larval stages [1], although this work did not identify the pro-

proliferative factors responsible for driving the phenotype. Remarkably, the proliferation effect seen in *Drosophila* tissue upon Dis3L2 knockdown is similar to that seen in human tissues suggesting a conserved role of Dis3L2 in maintaining control over cell proliferation and tissue growth. Here we build upon our previous work using CRISPR-Cas9 generated null mutants and have, for the first time, identified the RNA target of Dis3L2 in *Drosophila* that is responsible for driving the overgrowth phenotype. We demonstrate that loss of Dis3L2 results in the upregulation of a conserved growth factor named Imaginal Disc Growth Factor 2 (Idgf2). Idgf2 is a member of the conserved chitinase-like protein (CLP) family including CHI3L1/ YKL-40 and CHI3L2/YKL-39 which are implicated in inflammatory diseases and a variety of cancers including colon and non-small cell lung cancer. Finally, we demonstrate in both *Drosophila* and human HEK-293T kidney cells that overgrowth observed in Dis3L2 deficient tissues is wortmannin sensitive suggesting a novel and highly conserved mechanism whereby Dis3L2 controls proliferation and tissue growth potentially via the PI3-Kinase pathway.

## Results

### Dis3L2 null mutants show widespread tissue overgrowth

Our previous work, using RNA interference, demonstrated that *Drosophila* are an excellent model organism for the study of how Dis3L2 functions to control cell proliferation and tissue growth [1]. Since RNA interference has its limitations due to residual activity, we used CRISPR-Cas9 to generate a *dis3L2* null mutant as an optimal genetic model to understand the molecular pathways regulated by Dis3L2 to control proliferation. Using this approach, we produced a *dis3L2* mutant line with an 8bp deletion (*dis3L2$^{12}$*) in which no Dis3L2 protein is produced (Figs 1A, 1B and S1A). In parallel, we also generated an isogenic control line which remained unedited or was repaired correctly (*dis3L2$^{wt}$* (Fig 1A and 1B)). Crucially, *dis3L2$^{12}$* flies showed wing overgrowth similar to that previously observed in our RNAi flies (Fig 1C). To genetically confirm the null nature of the allele we created hemizygous flies by crossing the mutant *dis3L2$^{12}$* chromosome over the smallest available deletion that includes the *dis3L2* locus (*DF(3L)Exel6084*). Whilst hemizygous flies containing a single wild-type copy of Dis3L2 (*dis3L2$^{wt}$/Df*) showed no phenotype those containing the mutant chromosome (*dis3L2$^{12}$/Df*) showed significant wing overgrowth (13%) that was not different to homozygous *dis3L2$^{12}$* flies (Fig 1C). As with our previously published data the wing overgrowth is primarily driven by an increase in cell number in addition to a small (2%) increase in cell size (Fig 1D and 1D').

To determine the tissue specificity of *dis3L2* mutations, we checked for overgrowth of other tissues in our null mutants. These experiments showed that the overgrowth is not restricted to the wing but is also observed in the haltere and leg imaginal discs (Fig 1E). Phosphohistone H3 staining confirmed the increase in *dis3L2$^{12}$* tissue size is driven by an increase in proliferation in wing, haltere and leg imaginal discs (Fig 1F). Finally, although we observe a general increase in adult fly size (Fig 1G) we do not see a similar increase in size of *dis3L2$^{12}$* L3 larvae, or larval salivary glands suggesting the overgrowth effects are restricted to imaginal tissues (S1B Fig).

Increased growth would be expected to require increased protein translation. To assess the changes in global translation within the wing imaginal disc we used surface sensing of translation (SUnSET) labelling. SUnSET labelling uses puromycin as an analogue of tyrosyl-tRNA and therefore when cells are incubated with low amounts of puromycin it is added into the elongating peptide chain. The amount of incorporated puromycin over a given time is detected with a monoclonal antibody which allows the measurement of nascent translation. Using this approach, we confirmed an increase in translation within the wing imaginal discs (Fig 1H).

In addition to the overgrowth phenotype we were also able to confirm the male fertility phenotype observed by Lin and colleagues (S1C Fig) [11]. *dis3L2$^{12}$* male flies were completely

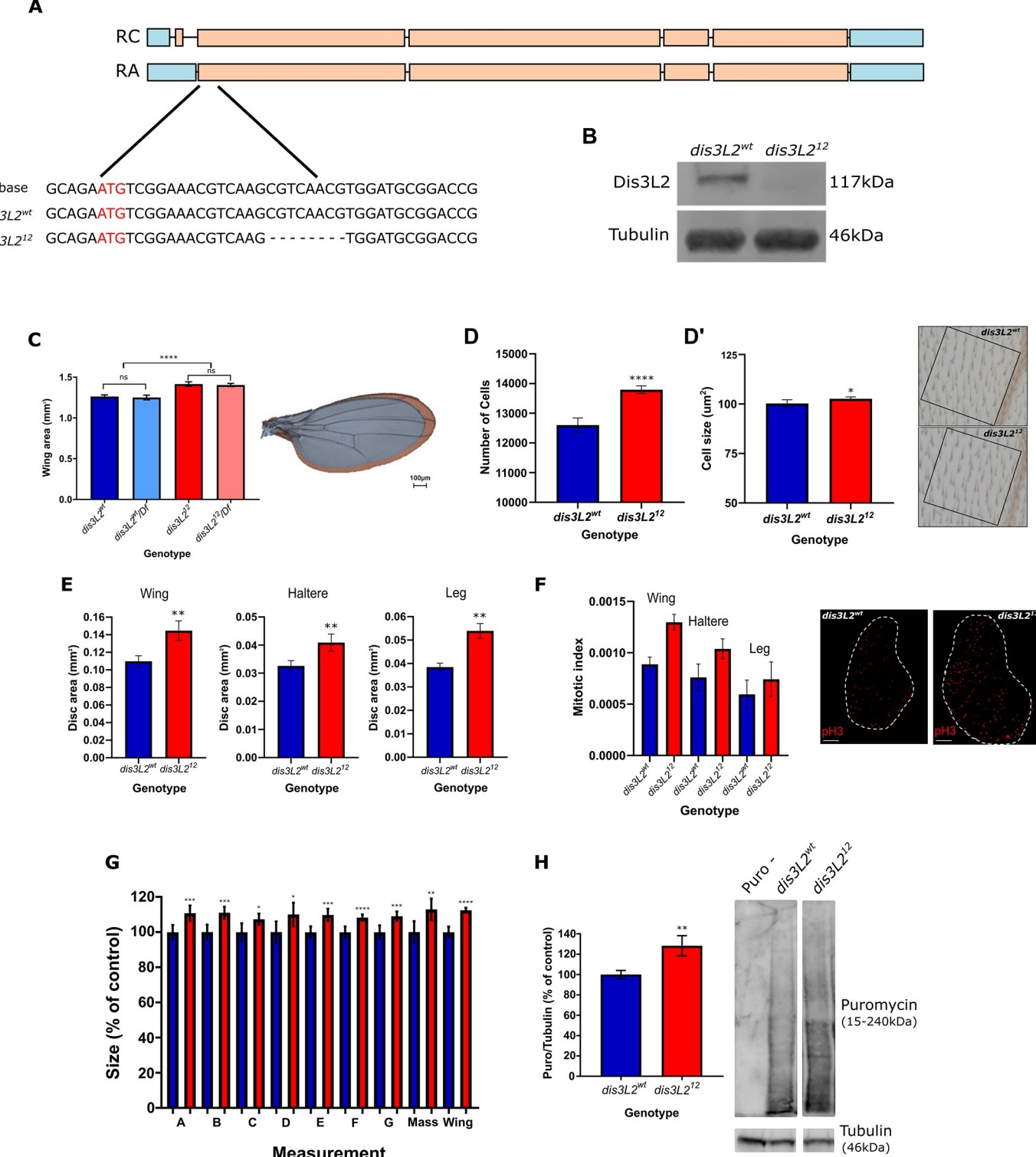

**Fig 1. *dis3L2^12* mutants show overgrowth phenotypes. A)** Schematic showing location and details of the *dis3L2^12* mutation. Coding exons are marked in orange and 5' and 3' UTRs in blue. RC and RA denote the two *dis3L2* isoforms and red letters denote the start codon. **B)** The 8bp mutation in *dis3L2^12* flies results in a complete absence of the Dis3L2 protein. **C)** *dis3L2^12* wings (red) are significantly larger (14%) than control *dis3L2^wt* wings (blue). Hemizygous *dis3L2^12/Df* wings are not significantly different from *dis3L2^12* wings demonstrating that it is a null mutation. n = 16–19, error bars represent 95% CI, **** = p<0.0001, ns = p>0.05, scale bar = 100μm. **D/D')** The increased wing area is mainly driven by an increase (12%) in cell number **(D)** and a small (2%) increase in cell size **(D')**. n = 13, error bars represent 95% CI, * = p<0.05, **** = p<0.0001. Representative images of one of 5 wing locations used to calculate wing area is shown for *dis3L2^wt* and *dis3L2^12*

wings. Each cell is marked by a trichome (hair) in the upper and lower cell layer. **E)** *dis3L2*$^{12}$ wing, haltere and leg imaginal discs are significantly larger (25%, 30%, 36% respectively) than their control counterparts. n = 8–33, error bars represent 95% CI, ** = p<0.01. **F)** Consistent with an increase in wing area, PH3 staining of wing, haltere and leg imaginal discs demonstrates an increased number of proliferating cells in *dis3L2*$^{12}$ tissues. n = 5–16, error bars represent 95% CI, scale bar = 50µm. **G)** *dis3L2*$^{12}$ mutants are significantly larger than *dis3L2*$^{wt}$ flies. Calculated using measurements of various regions of the adult fly (A-G, S7 Fig) and total mass. n = 15–21, error bars represent 95% CI, * = p<0.05, ** = p<0.01, *** = p<0.001, **** = p<0.0001. **H)** SUnSET labelling performed on wing imaginal disc samples shows increased global translation in *dis3L2*$^{12}$ discs compared to *dis3L2*$^{wt}$. Representative lanes from the same blot are shown along with Tubulin loading control used for quantification. Puro–represents a disc sample incubated in media only without Puromycin. n = 8–14, error bars represent 95% CI, ** = p = 0.0057.

infertile with seminal vesicles devoid of sperm. We further confirmed the specificity of the infertility phenotype in male flies where *dis3L2* had been ubiquitously knocked down using *Tubulin-GAL4* and *UAS-dis3L2*$^{RNAi}$ (S1C Fig). Finally, both male and female *dis3L2*$^{12}$ show a decreased lifespan compared to *dis3L2*$^{wt}$ flies (S1D Fig).

## Dis3L2 catalytic activity is required to maintain control of tissue growth

To confirm that loss of Dis3L2 is indeed driving the overgrowth phenotype we generated a number of transgenic lines to allow specific genetic rescue. There are two annotated isoforms of Dis3L2 which differ through the use of an alternative start codon resulting in two protein products of 1032 amino acids (isoform PA) and 1044 amino acids (isoform PC). To ensure that re-expression of Dis3L2 was to a level comparable to wild-type we used the site-specific recombination attB/attP system to allow the specific insertion of Dis3L2 cDNA into a genetic region which would result in mild expression. This mild expression was confirmed by Western blotting where driving ubiquitous ectopic expression of these lines using *actin5C-GAL4* resulted in a mild ~2 fold overexpression (Figs 2A and S2A) in *dis3L2*$^{wt}$ flies. Re-expression of Dis3L2 protein throughout *dis3L2*$^{12}$ mutant flies was achieved to 143% and 170% the level of control flies for PA and PC respectively (Compare -/PA and -/PC to +/- in Fig 2A and see S2A Fig for quantification). Expression of both Dis3L2 isoforms (PC) or only PA specifically in the cells fated to form the wing (using *nub-GAL4*) results in a complete rescue of wing area to that of *dis3L2*$^{wt}$ wings whilst the fly mass remained significantly larger (Figs 2B and S2B). To confirm the specificity of the phenotypic rescue we extended these findings using the *engrailed-GAL4* (*en-GAL4*) driver to allow specific re-expression of Dis3L2 in the posterior compartment of the wing whilst the anterior area remained mutant for Dis3L2, therefore acting as an internal control. This revealed a specific rescue of the posterior area in *en-GAL4* driven rescue wings whilst the anterior area remaining significantly larger (S2C Fig). Importantly, driving mild overexpression of Dis3L2 with these UAS lines with either *nub-GAL4* or *en-GAL4* has little effect on wing area alone (S2D and S2E Fig). These phenotypic rescues show that the loss of Dis3L2 is indeed responsible for the overgrowth phenotype and that isoform PA is the major isoform in *Drosophila* to regulate tissue growth.

Next, we asked if the catalytic activity of Dis3L2 was required to control proliferation by creating a nuclease dead transgenic line (*UAS-dis3L2*$^{ND}$) carrying a single amino acid substitution (D580N) which has previously been shown to abolish Dis3L2 catalytic activity [13]. To create this line, we used the same targeted insertion site which resulted in expression comparable to the wild-type constructs (Figs 2A and S2A). However, unlike the previous rescue experiments, specific expression of *dis3L2*$^{ND}$ results in no change in *dis3L2*$^{12}$ wing area when driven by either *nub-GAL4* (Fig 2B) or *en-GAL4* (S2C Fig) demonstrating the catalytic activity of Dis3L2 is indeed essential to control proliferation in the wing imaginal disc. Finally, to assess functional homology between the *Drosophila* and human Dis3L2 proteins we also created a line to allow us to express human DIS3L2 (Ensembl isoform 202, 885aa) in our *dis3L2*$^{12}$ flies (S2F Fig). Whilst expression of human DIS3L2 in a *dis3L2*$^{12}$ mutant background did not

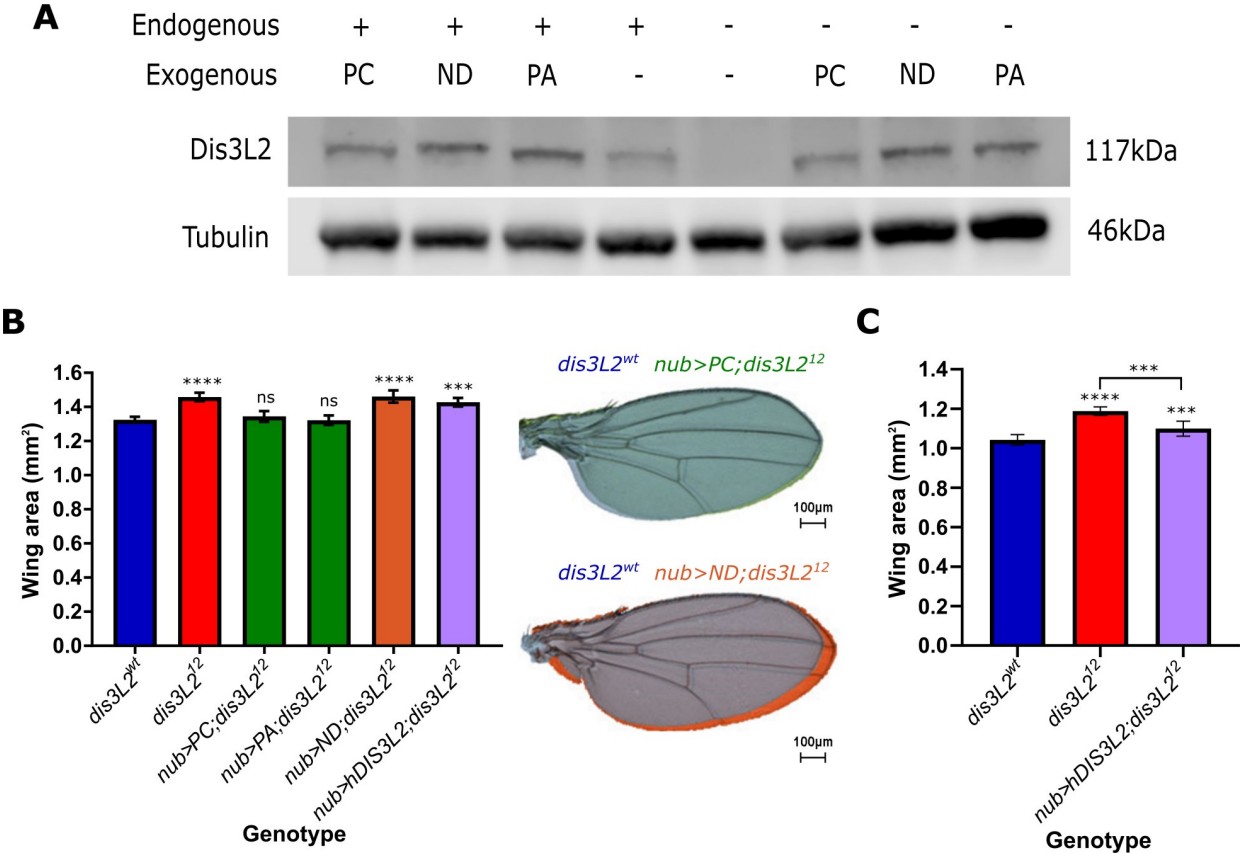

**Fig 2. Ectopic expression of catalytically active *dis3L2* rescues the wing overgrowth phenotype. A)** *UAS-dis3L2*[PA], *UAS-dis3L2*[PC], or *UAS-dis3L2*[ND] can be used to express Dis3L2 in *dis3L2*[12] animals. UAS constructs were driven in either *dis3L2*[wt] (+) or *dis3L2*[12] (-) animals with the ubiquitous driver *act-GAL4* with Western blots performed on whole female flies. See S2A Fig for quantification. **B)** Expression of both Dis3L2 isoforms (PC) or only the shorter isoform (PA) in the wing pouch of the wing imaginal disc is sufficient to completely rescue wing area in *dis3L2*[12] mutants. Expression of a catalytically dead Dis3L2 (ND) does not rescue the phenotype, whilst expression of human DIS3L2 (hDIS3L2) also shows no significant rescue at 25˚C. n = 14–65, error bars represent 95% CI, statistics shown refer to a comparison between the test genotype and control **** = $p < 0.0001$, *** = $p < 0.001$, ns = $p > 0.05$. **C)** When driven at 29˚C the human DIS3L2 construct (hDIS3L2) shows a partial (~50%) rescue of wing area. n = 16–28, error bars represent 95% CI, **** = $p < 0.0001$, *** = $p < 0.001$.

rescue wing overgrowth at 25˚C (Fig 2B), increasing the temperature to 29˚C resulted in a 50% rescue of wing area towards that of *dis3L2*[wt] flies also reared at 29˚C (Fig 2C). The requirement of an increased temperature may be due to the fact that the human protein would function optimally at 37˚C. The difference in rescue may also be due to the increased efficiency of the GAL4 system at a higher temperature and therefore more human protein would be produced. These data therefore demonstrate the catalytic activity of Dis3L2 is required to control proliferation and tissue growth and that there is at least partial functional homology between the human and *Drosophila* Dis3L2 proteins.

## Overexpression of Dis3L2 results in reduced proliferation

In addition to assessing the phenotypes consequent upon the loss of Dis3L2 we also investigated the *in-vivo* effect of strong overexpression of Dis3L2 using the wing imaginal disc as a model tissue. To achieve this, we used a publicly available line containing a P-element insertion, with an upstream activating sequence (UAS) 8bp upstream of Dis3L2 to achieve strong overexpression of the DIS3L2 protein (Fig 3A). Overexpression of Dis3L2 in the posterior

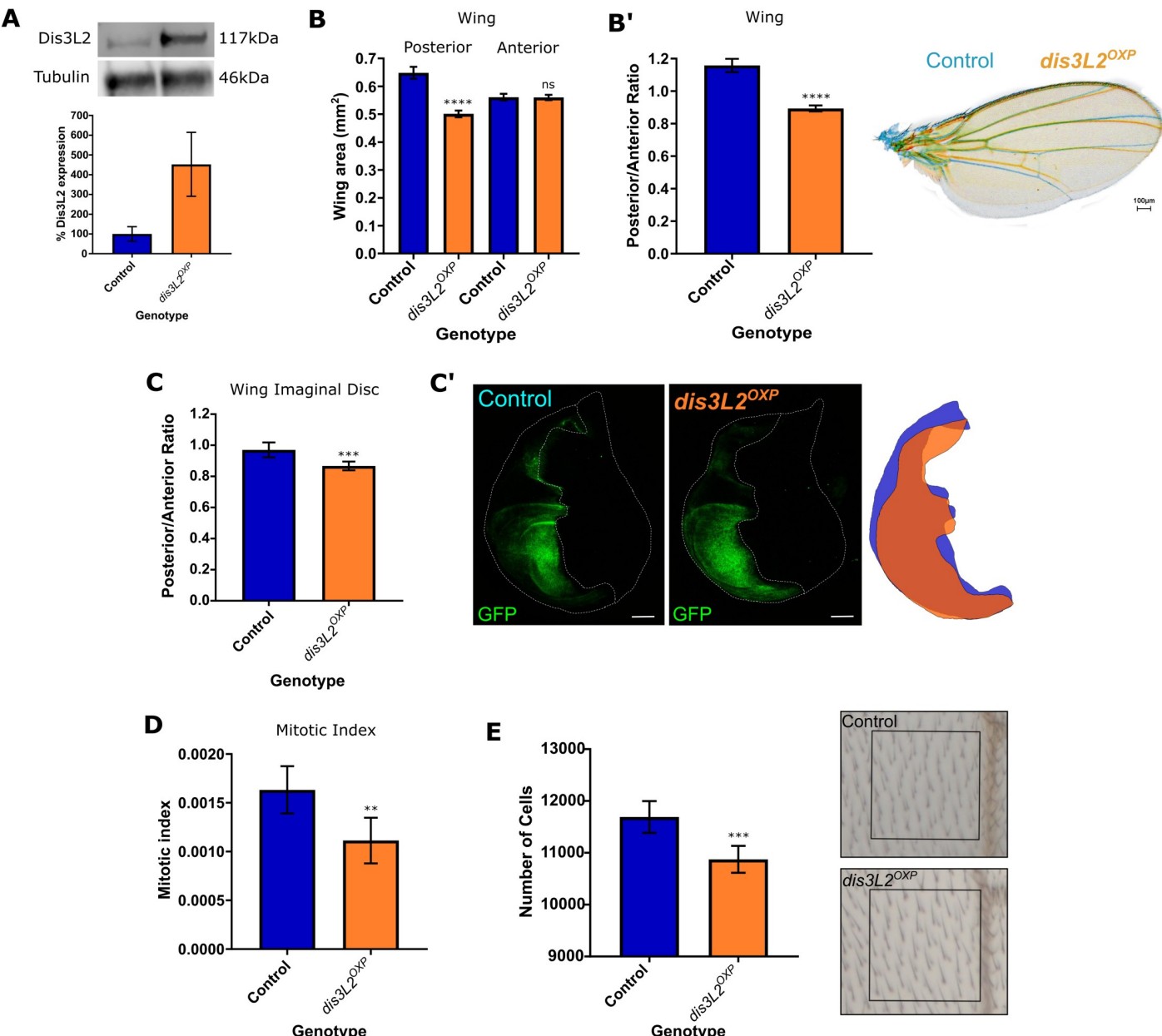

**Fig 3. Overexpression of Dis3L2 results in a reduction in proliferation in the wing imaginal disc. A)** Western blotting of *en-GAL4* driven Dis3L2 overexpression (*dis3L2$^{OXP}$*) wing imaginal discs confirms a 5-fold upregulation in Dis3L2 protein throughout the disc. Control consists of the parental controls with either the *UAS* or *GAL4* insertion alone, *UAS* parent shown. n = 3, error bars represent SEM. **B)** Overexpression of Dis3L2 using *en-GAL4* shows a significant reduction in the posterior area of the wing. Controls consist of parental controls to account for genetic background. n = 26–28, error bars represent 95% CI, **** = p<0.0001, ns = p>0.05. **B')** Normalisation of posterior area of the wing to the anterior area shows a significant reduction in the posterior area of the wing. n = 26–28, error bars represent 95% CI, **** = p<0.0001. Overlay or representative control (blue) and Dis3L2 overexpression (orange, *dis3L2$^{OXP}$*) wings also shown. **C)** Overexpression of Dis3L2 in the posterior compartment of the wing imaginal disc (using *en-GAL4*) results in a reduction of the posterior area, represented as the posterior area normalised to the size of the anterior area of the tissue. The control group consists of; *en-GAL4, UAS-GFPactin/+;* to allow measurement and ensure that the *en-GAL4* insertion and expression of GFP is taken into account. n = 26–33, error bars represent 95% CI, *** = p = 0.0008. **C')** Representative images of control and Dis3L2 overexpression (*dis3L2$^{OXP}$*) wing discs when *en-GAL4* is used to drive *UAS-GFP* expression and *UAS-Dis3L2* in the case of Dis3L2 overexpression, or just *UAS-GFP* in control tissues. Scale bar = 50μm. A cartoon overlay of posterior compartment of control (blue) and overexpression (orange) wing imaginal discs is also shown. **D)** Overexpression of Dis3L2 throughout the wing imaginal disc (using *69B-GAL4*) results in a reduction in the mitotic index of the wing imaginal disc. Control represents a combination of both parental controls (*;;69B-GAL4* and*;; UAS-Dis3L2*). n = 6–10, error bars represent 95% CI, ** = p = 0.0076. **E)** Consistent with the mitotic index calculations, overexpression of Dis3L2 throughout the wing imaginal disc using *69B-GAL4* results in a reduction in the total number of cells in the wing. n = 13–22, error bars represent 95% CI, ***p = 0.0005.

compartment of the wing imaginal discs (e*n-GAL4*) resulted in a significant and specific reduction in the area of posterior compartment of both the wing and the wing imaginal disc (20% and 13% respectively) (Fig 3B and 3C). In these experiments the anterior area, which functions as an ideal internal control, showed no change in area therefore demonstrating the specificity of the phenotype. Additionally, overexpression of Dis3L2 throughout the wing imaginal disc (using *69B-GAL4*) results in a significant reduction in proliferation, measured using phosphohistone H3 staining (Fig 3D), which in turn results in a reduction in the number of cells in the adult wing (Fig 3E). Taken together these data therefore confirm that Dis3L2 is a master regulator which plays a critical role in controlling developmental proliferation and ulti- mately tissue size without being limited by other accessory factors.

## RNA-sequencing reveals differential expression of pro-proliferative transcripts

As Dis3L2 regulates proliferation through its catalytic activity we hypothesised that it achieves this by degrading specific, growth promoting transcripts. To identify the RNAs that become misexpressed in *dis3L2*[12] wing discs, and that are responsible for driving the overgrowth phe- notype, we performed RNA-sequencing on 120hr old *dis3L2*[12] and control wing imaginal discs in triplicate following rRNA depletion (see Methods for sample details). RNA-sequenc- ing fastq files were then processed and aligned to the *Drosophila* genome (Flybase release 6.18; see methods for further details). Following differential expression analysis using individual replicate comparisons and stringent filtering criteria we identified 501 genes that showed dif- ferential expression in *dis3L2*[12] wing imaginal discs (Fig 4A). Of these, 344 were upregulated >1.34 fold and 157 were downregulated >1.34 fold (S3A Fig). The top 30 upregulated tran- scripts are shown in Fig 4B. Interestingly, transcripts differentially expressed in *dis3L2*[12] wing imaginal discs have a significantly shorter 3' UTR (446bp) than the global average (598bp) whilst the average lengths of the 5'UTR and CDS do not differ (S4B Fig) which is congruent with findings in *S. pombe* [3].

Consistent with previous findings we observed changes in expression in a variety of mRNAs and non-coding RNAs (ncRNAs) [2,11–14]. Regarding these ncRNAs that showed differential expression, we observed an increase in *RNaseMRP*:*RNA* including reads mapping to unprocessed *RNaseMRP*:*RNA* (S3B Fig) corroborating findings in *Drosophila* S2 cells and whole flies [13] suggesting that removal of unprocessed *RNaseMRP*:*RNA* is a global function of Dis3L2. This is also likely to be a conserved function as similar findings were observed in mouse embryonic fibroblasts [12]. Other ncRNAs found to be significantly upregulated include *asRNA*:*CR45208*, *CR40461* and *asRNA*:*CR42871*; the functions of these are at present unknown.

Gene ontology (GO) analysis of the upregulated protein-coding transcripts revealed an enrichment of genes involved in carbon metabolism, symporters and glycosidases; whilst 33% of downregulated transcripts are membrane proteins (S3C Fig). Remarkably, the top 20 upre- gulated transcripts include 3 imaginal disc growth factors (*idgf1/2/3*), characterised in the 'Gly- cosidase' GO term, which show 74.9, 10.3 and 6.6-fold increases respectively. In *Drosophila* there are six members of the Idgf family (Idgf1-6); of these only *idgf1*, *2* and *3* are sensitive to the *dis3L2* mutation with *idgf4*, *5* and *6* showing no changes in expression (S3D Fig) indicating specific regulation. In addition to the selected Idgfs, another mRNA encoding the conserved growth factor *miple1* increased in expression by 22.3-fold. In total 6 transcripts were selected for validation by qRT-PCR due to their known roles in cell proliferation as well as *syt4* which showed the largest increase in expression. qRT-PCR confirmed that all but *syt4* showed signifi- cant upregulation in both the samples sent for sequencing and in three additional replicates of

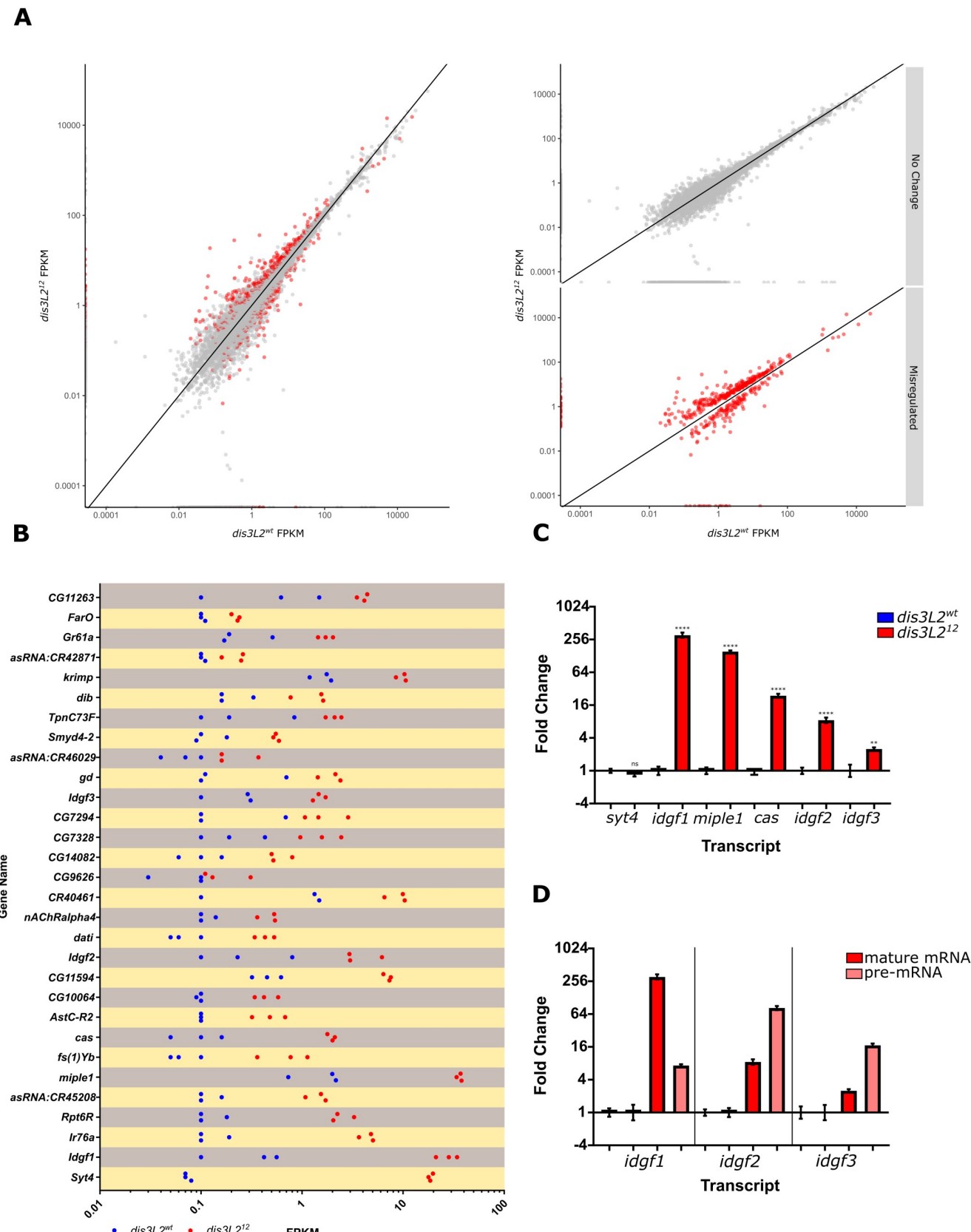

**Fig 4. RNA-seq analysis of transcripts misregulated in *dis3L2* mutants compared to controls. A)** 501 transcripts (red dots) were determined as differentially expressed between *dis3L2¹²* and control wing imaginal discs following stringent filtering criteria. **B)** Top 30 upregulated transcripts and their replicate FPKMs in control (blue) and *dis3L2¹²* (red) wing discs. **C)** 5 out of 6 selected transcripts showed significant increases in expression by qRT-PCR in both the sequenced samples and fresh validation samples. n = 6, error bars represent SEM, **** = p<0.0001, ** = p<0.01, ns = p>0.05. **D)** *idgf1, 2* and *3* are upregulated at the post-transcriptional as well as the transcriptional levels in the *dis3L2* null mutant compared to controls. qRT-PCR was used to detect the pre-mRNA and mRNA transcripts. n = 6, error bars represent SEM, p<0.05 for all.

*dis3L2¹²* and *dis3L2ʷᵗ* wing imaginal discs (Fig 4C). To confirm the upregulation was specific and not caused by an off-target effect of the CRISPR process we assessed the levels of the 5 upregulated transcripts (excluding *syt4*) in *dis3L2¹²* hemizygote wing discs. Significant upregulation of each transcript was also observed in these tissues, confirming their Dis3L2 sensitivity (S4C Fig).

## Idgf2 drives wing overgrowth in *dis3L2¹²* flies

In order to narrow down the mRNAs sensitive to *dis3L2* mutations, we used additional, publicly available data sets derived from different *Drosophila* tissues. Re-analysis of RNA-sequencing data from *dis3L2* null mutant testes [11] showed that 10% of transcripts differentially expressed in the mutant wing imaginal discs were also misexpressed in mutant testes (S4D Fig) showing that transcripts can be both globally sensitive to Dis3L2 or show tissue specificity. Upregulation of *idgf1, 2* and *3* transcripts but not *miple1* and *cas* was observed in both tissues suggesting intrinsic sensitivity to Dis3L2. If Dis3L2 is able to directly degrade *idgf1, 2* and *3* we would expect these RNAs to be bound by Dis3L2. Since CLIP-seq analysis, which can detect the binding of proteins to RNAs, is challenging to perform on the small amounts of imaginal disc tissue dissected from *Drosophila* larvae, we turned to an existing publicly available data from *Drosophila* S2 cells [13]. These data show that *idgf1, idgf2* and *idgf3* transcripts are all pulled down with a catalytic dead Dis3L2 in *Drosophila* S2 cells [13], demonstrating that they are physically associated with Dis3L2 and that Dis3L2 can directly target and degrade these transcripts. We also assessed the transcriptional contribution towards the increased levels of *idgf1, 2* and *3* by assessing changes in pre-mRNA expression. All three transcripts also showed pre-mRNA increases in the *dis3L2* mutant (Fig 4D), therefore it is likely that the increase observed in *dis3L2¹²* tissues is a consequence of a combination of transcriptional and post-transcriptional effects.

Since the transcripts encoding *idgf1, 2* and *3* appear to be globally sensitive to Dis3L2 and directly bound to the enzyme we used the power of *Drosophila* genetics to assess their individual contribution towards the overgrowth phenotype in *dis3L2¹²* animals. To achieve this, we used *UAS-RNAi* lines to *idgf1, 2* or *3* each of which showed a knockdown of >90% at the RNA level when driven ubiquitously (by *Tubulin-GAL4*) with no obvious phenotypic defects (S5A Fig). As in our previous rescue experiments, we again made use of the *en-GAL4* driver to deplete the levels of each *idgf* transcript individually in a *dis3L2¹²* mutant background. If any of these *idgfs* were individually responsible for the overgrowth phenotype we would expect individual knockdown to rescue the overgrowth phenotype resulting from the *dis3L2* null mutation specifically in the posterior region as is observed when *UAS-Dis3L2* is driven by *en-GAL4* (Figs 5A and S2C). Using this approach to knockdown *idgf1* or *idgf3* in the *dis3L2* mutant background had no effect on the posterior area of the tissue (Fig 5A). In contrast, depletion of *idgf2* resulted in a complete and specific rescue of the posterior area of the wing (Fig 5A). Importantly, knockdown of *idgf2* in a *dis3L2* wild-type background has no effect on the area of the tissue (Fig 5B). Consistent with a role for Dis3L2 in regulating Idgf2, re-expression of Dis3L2 specifically in *dis3L2¹²* mutant wing discs also results in a reduction of *idgf2* mRNA (S5B Fig). Further, levels of *idgf2* mRNA were also significantly increased in wing

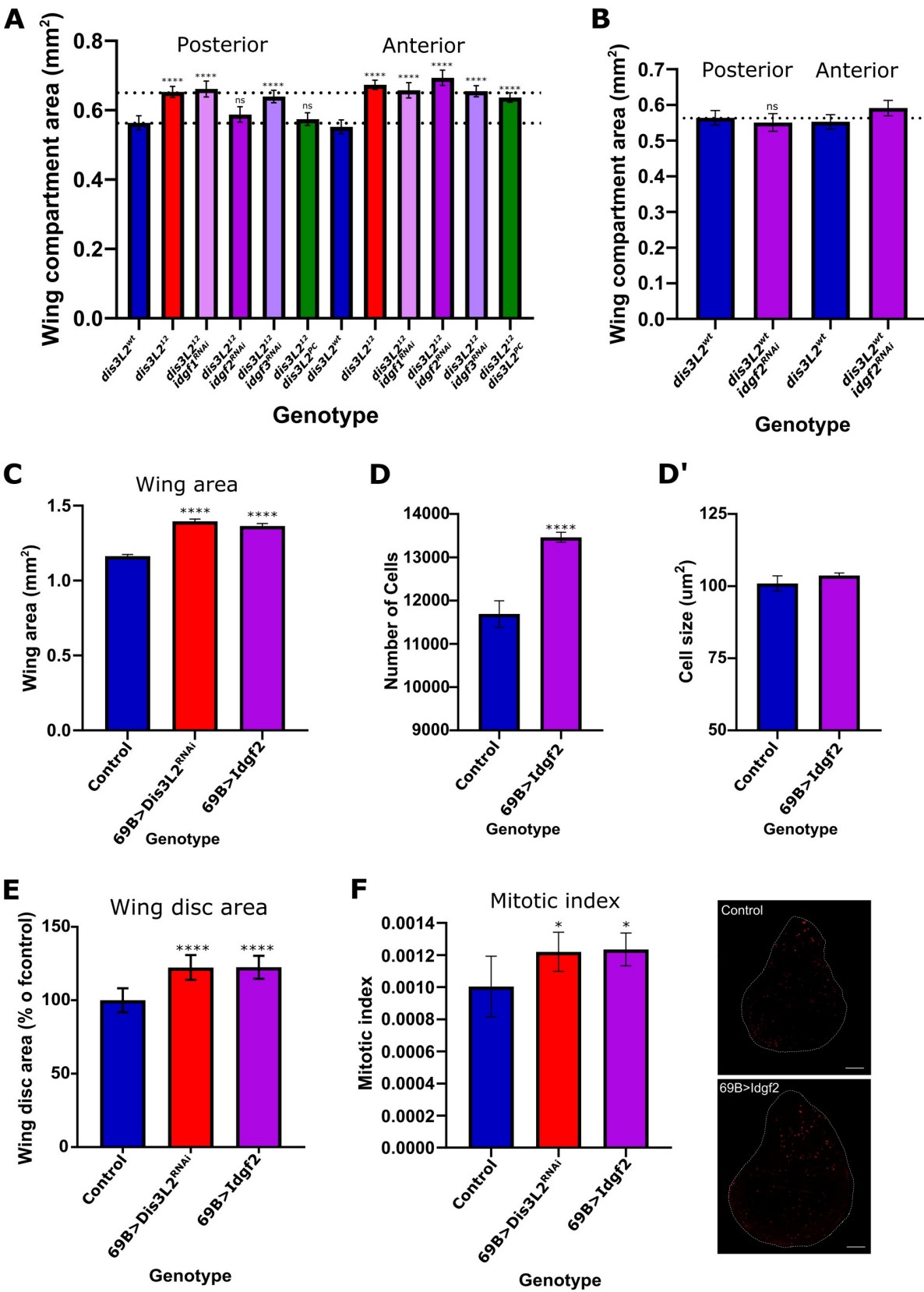

**Fig 5. Idgf2 drives tissue overgrowth in *dis3L2* mutant animals. A)** Knockdown of *idgf2* (purple) but not *idgf1* (pink) or *idgf3* (pale purple) in the posterior region (*en-GAL4*) of *dis3L2* mutant wing imaginal discs rescues the wing overgrowth phenotype to the same extent as expressing Dis3L2 itself (green). n = 16–34, error bars represent 95% CI, **** = p<0.0001, ns = p>0.05. Dotted lines represent the average area of the posterior region of mutant and control wings. **B)** Knockdown of *idgf2*, using *en-GAL4*, in a *dis3L2^{wt}* background has no effect on wing area. n = 16–18, error bars represent 95% CI, **** = p<0.0001, ns = p>0.05. **C)** Overexpression of Idgf2 throughout the wing imaginal disc using the *69B-GAL4* driver (purple) results in overgrowth of the wing to the same extent as observed when *dis3L2* is knocked down using the same method (red). n = 18–35, error bars represent 95% CI, **** = p<0.0001. **D/D')** Assessment of wing cell number (D) and wing cell size (D') when Idgf2 is overexpressed throughout the wing using *69B-GAL4* demonstrates an increase in cell number. n = 19–22, error bars represent 95% CI, **** = p<0.0001. **E)** Idgf2 overexpression (with *69B-GAL4*) results in an increase in wing imaginal disc area. n = 9–13, error bars represent 95% CI, ****p<0.0001. **F)** Consistent with an increase in cell number Idgf2 overexpression wing imaginal discs (*UAS-Idgf2/+; 69B-GAL4/+*) show increased proliferation to a level consistent with that observed in *dis3L2* deficient wing imaginal discs (*UAS-dis3L2^{RNAi}/+; 69B-GAL4/+*). n = 9–13, error bars represent 95% CI, * = p<0.05, scale bar = 50μm.

imaginal discs dissected from a publicly available independent line carrying a CRISPR engineered catalytic dead *dis3L2* mutation [13] (S5C Fig).

To confirm that increased expression of Idgf2 alone is sufficient to drive tissue overgrowth and proliferation *in-vivo* we generated a *UAS-Idgf2* transgenic line which resulted in a significant increase *idgf2* expression when expressed throughout the wing imaginal disc (using *69B-GAL4*) (S5D Fig). Overexpression of Idgf2 throughout the wing imaginal disc resulted in significant wing and wing imaginal disc overgrowth (as a result of increased proliferation in the wing imaginal disc and subsequent increased number of cells in the adult wing) to a level not different from that observed following *dis3L2* depletion using the same *69B-GAL4* driver (Fig 5C–5F).

## Loss of DIS3L2 in HEK-293T cells drives proliferation and upregulates the PI3-Kinase pathway

Given the lack of knowledge about the mechanism whereby Idgf2 drives proliferation in *Drosophila* we turned to human cell culture to identify the signalling pathway activated in DIS3L2 deficient cells. Previous work in HeLa cells demonstrated that reduction in DIS3L2 levels results in an increase in cell number [9]. Given the association between DIS3L2 and the kidney tumour, Wilms' Tumour, we tested whether the loss of DIS3L2 was sufficient to promote proliferation in HEK-293T cells, which are a more physiologically relevant, embryonic kidney derived cell line. Using siRNA against *DIS3L2* we achieved a maximal knockdown of 91.8% (72hrs post transfection) with a knockdown of >85% between 48 and 96 hours post transfection (144hrs post transfection a 62% knockdown was still present) (S6A Fig). During this period we observed an increase in cell number in the *DIS3L2* knockdown cells compared to a scrambled siRNA control, demonstrating a conserved role for controlling proliferation in a clinically relevant cell type (Fig 6A). Interestingly, the hyperplasia resulting from DIS3L2 depletion has some tissue specificity as the same experiments in the osteosarcoma cell line U-2 OS showed no phenotype (Figs 6B and S6B).

Having confirmed the conservation of the phenotype between *Drosophila* and human cells, we set out to identify the molecular pathway controlled by DIS3L2 to regulate proliferation. We first made use of publicly available RNA-sequencing data from HEK-293T cells that conditionally express a catalytically dead DIS3L2 [14] and identified 111 differentially expressed transcripts. Gene ontology analysis of these transcripts indicated the potential enrichment of the PI3 Kinase (PI3-K) pathway (the only significant hit with an enrichment score of 1.93). This would be consistent with previous data showing that ectopic expression of an Idgf2 orthologue, CHI3L1, activates the AKT pathway in a variety of human cells [18–20]. To test the effect of depletion of DIS3L2 on the PI3-K/AKT pathway we used Western blotting to assess the phosphorylation status of AKT (T308 and S473) in addition to downstream pathway

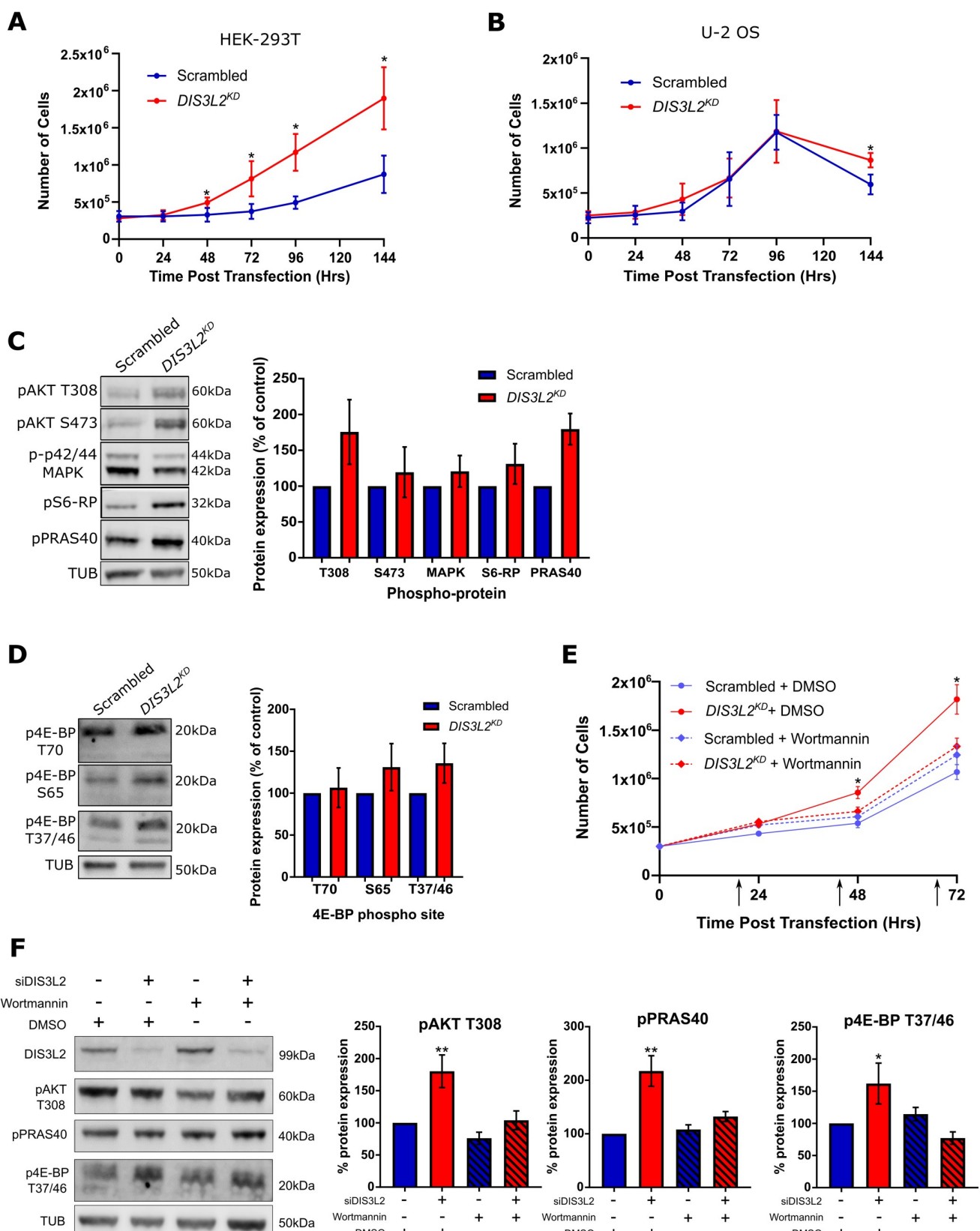

**Fig 6. DIS3L2 depletion in HEK-293T cells results in PI3-K induced hyperplasia. A)** Knockdown of *DIS3L2* in HEK-293T cells confers a growth advantage compared to a scrambled control. n = 6, error bars represent 95% CI, * represent significant differences where error bars to not overlap. **B)** Knockdown of *DIS3L2* in U-2 OS cells has no effect on cell number compared to a scrambled control. n = 6, error bars represent 95% CI, * represent significant differences where error bars to not overlap. **C)** Representative image and quantification of key growth promoting phospho-proteins in HEK-293T cells treated with siDIS3L2 (*DIS3L2^{KD}*) or siScrambled (Scrambled). n = 5, error bars represent SEM. **D)** Representative image and quantification of three different 4E-BP phosphorylation sites in HEK-293T cells treated with siDIS3L2 (*DIS3L2^{KD}*) or siScrambled (Scrambled). n = 3, error bars represent SEM. **E)** Treatment of *DIS3L2^{KD}* HEK-293T cells with Wortmannin (250nM) results in a rescue of the proliferation phenotype to that of a scrambled control with or without Wortmannin treatment (DMSO). Note that the growth of the scrambled control is not affected by this dose of Wortmannin. n = 4, error bars represent 95% CI, * represent significant differences where error bars to not overlap. **F)** Western blot showing the increased levels of pAKT (T308), pPRAS40 or p4E-BP (T37/46) in *DIS3L2^{KD}* HEK-293T cells 72hrs post transfection is lost upon wortmannin treatment. Hatched bars represent samples treated with wortmannin, n = 5, error bars represent SEM. ** = p<0.01, * = p<0.05.

members. We also assessed the phosphorylation status of p42/44 MAPK, another crucial, conserved growth mediator also shown to be activated by ectopic CHI3L1/2 expression [21–23]. This demonstrated that loss of DIS3L2 did not affect MAPK phosphorylation but results in increased phosphorylation of pAKT at T308, and a complementary increase in phosphorylation of one of its targets, pPRAS40 (Fig 6C). T308 is known to be phosphorylated by PDK1 following PI3-K activation [24], whereas S473, which shows a partial but inconsistent increase, is phosphorylated by mTORC2 [25].

Having identified an apparent activation of PI3-K/AKT signalling in DIS3L2 deficient cells we asked if mTORC1 was activated, as might be expected for PI3-K dependent growth. We assessed the phosphorylation status of the mTORC1 target, 4E-BP on 4 conserved sites. This approach demonstrated that increased phosphorylation on S65 and T37/46 was observed in *DIS3L2* knockdown cells (Fig 6D). However, we could not confirm increased T70 phosphorylation. To ask if the observed changes in phosphorylation status represented post-translational pathway activation or protein upregulation, we assessed changes in total levels of AKT, PRAS40 and 4E-BP. In all cases we saw changes in phosphorylation were over and above changes in steady state levels of the proteins (S6C Fig). However, due to some technical variation it is unclear as to whether the changes are solely driven by increased phosphorylation and it remains possible that these observations could be caused a combination of protein upregulation and post-translational modification. To phenotypically assess if the PI3-K pathway is responsible for driving overgrowth in DIS3L2 knockdown HEK-293T cells we treated them with the PI3-K inhibitor wortmannin. Whilst the addition of 250nM wortmannin to control cells did not appear to affect growth, wortmannin treatment of *DIS3L2* knockdown HEK-293T cells resulted in a complete rescue of overgrowth, demonstrating a growth profile similar to that of the scrambled control with or without wortmannin (Fig 6E). Wortmannin activity was confirmed by assessing the levels of pAKT (T308), pPRAS40 and p4E-BP (T37/46) by Western blot which showed a specific reduction in phosphorylation in *DIS3L2^{KD}* HEK-293T cells following wortmannin treatment (Figs 6F and S6D). Taken together, although further work is required to elucidate the manner of PI3-Kinase pathway upregulation, this data demonstrates the first example of a mechanism through which DIS3L2 regulates proliferation in human cells, and may be consistent with the overgrowth induced by the orthologues of Idgf2.

## Increased proliferation in *dis3L2^{12}* wings is wortmannin-sensitive

Having identified a role for the PI3-K/AKT pathway in *DIS3L2* knockdown HEK-293T cells we asked if the same pathway was mediating growth in the *Drosophila* wing imaginal discs. To address this, we fed wortmannin to larvae at concentrations of 5μM and 10μM at various stages during their development. At these optimised concentrations, the wild-type larvae are not affected. Whilst the presence of wortmannin from 24, 48 or 72hrs into development had no effect on *dis3L2^{wt}* wing area, *dis3L2^{12}* wings were significantly smaller than those fed the

vehicle control food (containing DMSO) (Fig 7A–7C). Although the rescue of wing area was not complete, we saw a significant reduction in the overgrowth suggesting that Dis3L2-mediated control of growth occurs in a conserved manner at least partially through a PI3-Kinase sensitive network. Consistent with a role for Idgf2 in driving the overgrowth phenotype in *dis3L2*$^{12}$ animals, the feeding of wortmannin to flies overexpressing Idgf2 (throughout the wing imaginal disc using *69B-GAL4*) also results in a comparable reduction in tissue area (Fig 7D). Additionally, and congruent with our HEK-293T cell data, we observe increased phosphorylation of AKT at T342 (homologous to T308) in both *dis3L2*$^{12}$ larvae and those with ubiquitous Idgf2 overexpression demonstrating the conservation of Dis3L2-mediated growth control (Fig 7E). However, it is important to note that we were unable to assess total AKT levels in our mutant/overexpression animals and therefore further work is required to identify the mechanism behind the increased phosphorylation of AKT. Finally, although wing area is reduced in *dis3L2*$^{12}$ flies following wortmannin treatment, the addition of wortmannin has no effect on the overexpression of *idgf2* within the wing imaginal disc showing that Idgf2 overexpression lies genetically upstream of the activation of the PI3-K pathway in these tissues (Fig 7F). Together these data are consistent with a role for Idgf2 in driving the wortmannin sensitive pathway to promote proliferation and tissue overgrowth in *dis3L2*$^{12}$ tissues. Further work is required to investigate if Idgf2 directly activates the PI3-Kinase network in *Drosophila* together with the nature of the pathway activation; the identification of the Idgf2 receptor is crucial to this concept. Nevertheless, we have, for the first time, uncovered a conserved mechanistic pathway whereby Dis3L2 regulates cell proliferation in both *Drosophila* and human cells.

## Discussion

### Dis3L2 induced overgrowth in *Drosophila* requires upregulation of the growth factor Idgf2

In this study we have used a new *dis3L2* null mutant to identify Idgf2 as the factor responsible for driving proliferation and overgrowth in *dis3L2* loss of function *Drosophila* tissues. Using RNA-sequencing data generated in this study, together with our analysis of published data, it is clear that *idgf2* expression is sensitive to Dis3L2 in a variety of *Drosophila* tissues. Crucially, *idgf2* mRNA has been shown to co-precipitate with a catalytic dead Dis3L2 in *Drosophila* embryonic S2 cells, suggesting Dis3L2 is able to directly degrade *idgf2* mRNA. Interestingly, *idgf2* mRNA also increases in levels, although to a lesser extent, in testes mutant for the terminal uridylyl transferase *tailor* [11], indicating that *idgf2* may be targeted to Dis3L2 by 3' uridylation. Further, re-expression of Dis3L2 specifically in the wing disc results in specific reduction in *idgf2*. Therefore, although further work is required to assess the ways in which Dis3L2 mediates clearance of *idgf2* transcripts, these analyses provide strong evidence that Dis3L2 functions to control proliferation in *Drosophila* by regulating *idgf2* expression. Interestingly, we also observe an increase in *idgf2* pre-mRNA suggesting an additional increase at the transcriptional level. This therefore suggests that the increase in *idgf2* in *dis3L2* mutant tissues is a combination of direct and indirect effects which subsequently results in the overgrowth of the tissue.

Although *Drosophila* Idgf2 is the best molecularly characterised of the Idgfs in that its crystal structure has been determined [26], the downstream pathway it activates remains elusive, although data presented here suggests that Idgf2 could promote PI3-K signalling to drive proliferation. For example, the fact that both Idgf2 overexpression and Dis3L2 mutant larvae show increased phosphorylation of AKT at T342, demonstrate wortmannin-sensitive overgrowth, and that *idgf2* expression remains elevated in *dis3L2* mutant tissues treated with

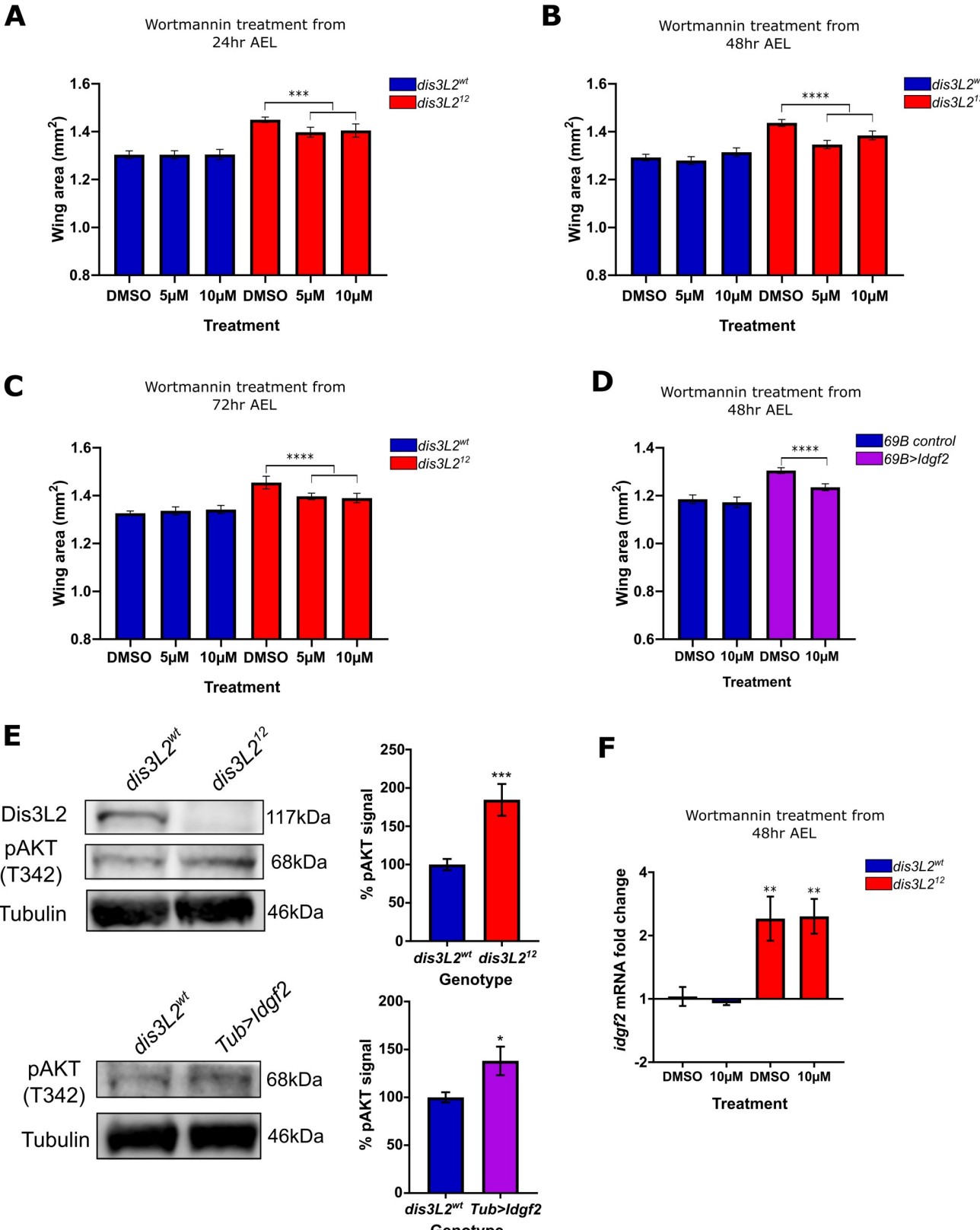

**Fig 7. PI3-K is involved in driving proliferation in *dis3L2¹²* mutant wings. A-C)** Feeding of Wortmannin to *Drosophila* larvae at 5µM or 10µM results in a significant reduction specifically in *dis3L2¹²* wings and not *dis3L2ʷᵗ* wings. Developing larvae were cultured on Wortmannin or control (DMSO)

containing food from **(A)** 24, **(B)** 48 or **(C)** 72 hours after egg lay (AEL). n = 16–58, error bars represent 95% CI, *** = p<0.001, **** = p<0.0001. **D)** Feeding of Wortmannin to larvae at 10μM results in a significant reduction of Idgf2 overexpression (*69B>idgf2*) wing area but not control (*69B-GAL4* parental) wings. Developing larvae were cultured on Wortmannin or control (DMSO) containing food from 48 hours AEL. n = 20–33, error bars represent 95% CI, **** = p<0.0001. **E)** Assessment of AKT phosphorylation at T342 in *dis3L2^wt^*, *dis3L2^12^*, and Idgf2 overexpression (driven by *Tub-GAL4*) 120hr old L3 larvae shows increased T342 phosphorylation in L3 larvae mutant for *dis3L2* and those with ubiquitous Idgf2 overexpression. n = 6–10, error bars represent SEM, * = p = 0.0121, *** = p = 0.0003. **F)** Addition of 10μM Wortmannin does not affect *idgf2* overexpression in *dis3L2^12^* wing imaginal discs showing Idgf2 is upstream of the Wortmannin sensitive pathway. n = 3–4, error bars represent SEM, ** = p<0.01.

wortmannin suggests that Idgf2 lies upstream of the wortmannin-sensitive pathway. We have shown that increased Idgf2 is sufficient to drive proliferation and tissue growth and results in increased AKT phosphorylation at T342 which provides *in-vivo* evidence to support previous findings that Idgf2 promotes proliferation of *Drosophila* wing disc cells in culture [27,28]. Whilst the wortmannin data suggests the wing overgrowth is indeed PI3-K sensitive it is important to note that due to technical difficulties regarding the small size of the wing imaginal discs these preliminary conclusions are drawn on system-wide changes across the larvae. Further work would therefore be required to identify of the nature of increased pathway activity in addition to specifically addressing the signalling mechanisms within the wing imaginal disc. Interestingly, Blanco *et al* showed that Idgf2 increases in expression by 4-fold during early stages of tissue regeneration [29]. Further, it has recently been shown that Idgf2 protects imaginal disc cells (C1.8) in culture from apoptosis and that this cytoprotection is associated with the induction of genes associated with energy metabolism and innate immunity [27]. Our SUnSET labelling, which shows increased translation in *dis3L2* mutant cells, would support the idea that these cells are more physiologically active than normal cells. It is therefore possible, that under normal conditions, Dis3L2 functions to restrain Idgf2 signalling, however, in its absence the wing disc cells are 'primed' in a more proliferative state to facilitate the repair of damaged tissue or protect the cells from death.

Idgf2 is a member of the conserved Chitinase-like protein (CLP) family and orthologous to CHI3L1 and CHI3L2. Consistent with our findings, both CHI3L1 and CHI3L2 have been shown to drive proliferation in a variety of human cell types [18,20–23,30] in addition to being associated with numerous chronic inflammatory diseases and cancers including glioblastoma, non-small cell lung cancer and colon cancer. Interestingly, in the case of colorectal cancer, downregulation of DIS3L2 (as a consequence of knockdown of lncRNA *AC105461.1*) enhances the proliferation and stem-cell like properties of cells of the colorectal cancer line SW480 [31]. In line with the data presented here, ectopic CHI3L1 expression has been shown to stimulate AKT phosphorylation in a variety of cell types [18–20,30,32]. However, CHI3L1 and CHI3L2 are expressed at very low levels in immortalised cell lines and we were unable to reliably detect an increase in CHI3L1 or CHI3L2 transcripts in our HEK-293T cells. Therefore, it is also possible that another factor may be responsible for the hyperplasia observed in our DIS3L2-deficient HEK-293T cells.

## DIS3L2 represses PI3-K/AKT pathway activity in HEK-293T cells

DIS3L2 loss of function mutations have been implicated Perlman Syndrome and Wilms' tumour of the kidney [9]. To assess the role of DIS3L2 in human kidney cells we depleted DIS3L2 in HEK-293T cells and, as in *Drosophila*, we observed strong increases in proliferation and identified a potential role for the PI3-Kinase in driving this phenotype. The role of DIS3L2 in human cells was also shown to have some specificity since the overgrowth phenotype is not observed in following DIS3L2 knockdown in the osteosarcoma cell line U-2 OS. Although the PI3-K pathway is active in both these cell lines, there are many possible reasons why the two cell lines do not behave in the same way. For example, the differences could also be due to the

tissue of origin of the cell line, or could be due to HEK-293T being virally immortalised rather than cancerous and therefore this genetic difference may also be the cause of the different behavior of the two cell lines. Mammalian target of rapamycin (mTOR), is a downstream mediator in the PI3-K signalling pathway and is an essential serine/threonine kinase which is aberrantly activated in human osteosarcoma [33,34]. As this activation of mTORC is often independent of upstream signalling, including PI3-K, this could provide an explanation for the lack of influence of DIS3L2 on proliferation in osteosarcoma cells. Alternatively, given that Perlman Syndrome and Wilms' tumour are early, developmental diseases is could be that DIS3L2 is essential to maintain proliferation early in development. U-2 OS cells are from an adolescent whilst HEK-293T cells are embryonic and therefore this could influence the effect of loss of DIS3L2. This would be congruent with our previous work showing that Dis3L2 is critical to control proliferation during early development in *Drosophila* [1].

We observed increases in AKT/mTORC1 activity following the loss of DIS3L2 in HEK-293T cells, providing the first mechanistic explanation for DIS3L2 induced overgrowth in mammalian cells, although further work is required to better understand that nature of pathway activation. Interestingly, we saw increased phosphorylation of the PDK1 sensitive T308 site of AKT, however, we did not see consistent increases in S473 phosphorylation, catalysed by mTORC2 [25]. T308 has been shown to be sufficient for AKT activity, although its activity is reduced without S473 phosphorylation [35,36]. Consistent with our data, in both *Drosophila* and mouse models it has been shown that phosphorylation of the hydrophobic motif (S473 in humans and mice, S505 in *Drosophila*) is important for FOXO regulation but is not required for mTOR activation and regulation of T342 phosphorylation in *Drosophila* is important for growth control [37–40]. Whilst the lack of S473 activation may due to technical issues, it may also explain the controlled tissue growth we see in *Drosophila*. For example, the partial activation of AKT may be sufficient to drive increased growth through TORC1, however, without S473 activation, and subsequent FOXO inhibition, some control over proliferation is retained. Whilst we observe increased phosphorylation of AKT at T342 *in-vivo* in both Dis3L2 null and Idgf2 overexpressing animals, the lack of S473 phosphorylation in our experiments in HEK-293T cells is also consistent with the lack of activation of the corresponding site in *Drosophila* (S505) following ectopic Idgf2 expression in C1.8 cells [27].

Our conclusion that the PI3-K pathway contributes towards loss of DIS3L2 induced overgrowth is strengthened by the observation that the increased proliferation in DIS3L2 deficient cells was specifically inhibited following wortmannin treatment. These findings are also congruent with our re-analysis of RNA-seq data from DIS3L2 catalytic dead HEK-293T cells [14] which, in our hands, shows the only significantly differentially expressed pathway to be the PI3-K pathway. This apparent role for PI3-K in mediating Dis3L2-regulated growth appears to be conserved as we also present preliminary data in our *Drosophila* model demonstrating wortmannin-sensitivity and increased AKT phosphorylation. The mechanism behind the increased phosphorylation and whether Idgf2 directly activates this pathway, or there is co-operation with other pathways in *Drosophila* requires further work and remains under investigation.

## How is Dis3L2 recruited to its RNA targets?

At present the molecular mechanisms whereby Dis3L2 specifically regulates the expression of *idgf2* are unknown. One hypothesis is that sequences or structures within the mRNA itself render it particularly susceptible to degradation by Dis3L2. Previous work in human and *Drosophila* cells has identified a role for Dis3L2 in regulating RNA polymerase III transcripts [12–14] which have U-rich 3' ends as a result of pol III terminators. Interestingly, MEME [41] analysis

of the 3' UTRs of mRNAs upregulated in $dis3L2^{12}$ mutant wing discs which also co-precipitate with Dis3L2 [13] show a significant enrichment of a U-rich motif (40% submitted transcripts), and a CA-rich motif (14.1% of submitted transcripts) which are not enriched in a control data set (mRNAs that do not change in expression or co-precipitate with Dis3L2 in S2 cells) (S4E Fig). Given the preference of Dis3L2 in degrading U-rich regions this could demonstrate an intrinsic mechanism facilitating Dis3L2-mediated decay of a specific set of mRNA substrates. Further, the control data set showed enrichment of a G-rich motif which was not identified in the list of likely Dis3L2 substrates (S4F Fig). The inability of Dis3L2 to degrade G terminated RNAs [13], together with the fact that G-rich regions form stabilising G-quadruplex structures further strengthens the validity of our target dataset. Finally, transcripts with shorter 3'UTRs appear to be more sensitive to Dis3L2 activity (446nt vs 598nt; S4B Fig) which is consistent with data from *S. pombe* [3], however, the reason behind this remains unknown. Consistent with these findings, *idgf2* has a very short 3' UTR at 50bp in length and although it does not contain an extended U-rich motif as identified above, it does contain an AU-rich element which has been shown to target transcripts for 3'-5' decay by both the exosome and Dis3L2.

Whilst the data presented here demonstrate a novel, and potentially conserved, mechanism through which Dis3L2 mediated decay of *idgf2* is required to control tissue growth, the downstream signalling pathway activated by Idgf2 remains elusive. However, it appears that the involvement of the PI3-K pathway in inducing Dis3L2-mediated overgrowth is conserved between *Drosophila* and human HEK-293T cells given our identification of increased AKT phosphorylation in both models. Further understanding of these pathways is critical to understanding not only the DIS3L2 regulated conditions but the many other conditions also driven by the increased expression of the CLP family of proteins. Exploring the endogenous role of CLP proteins in tissue culture cells is difficult due to their low levels of expression and therefore our *in-vivo Drosophila* model is an exciting and appropriate system to further unlock this largely unknown pathway of tissue growth and proliferation.

## Methods

### Drosophila husbandry

Fly stocks were cultivated on standard media in uncrowded conditions at 25°C unless otherwise stated. The following stocks were obtained from Bloomington Stock Center: *Tub-GAL4* ($P\{w^{+mC} = tubP\text{-}GAL4\}LL7$ originally from stock 5138, $y^1 w^*$;; $P\{w^{+mC} = tubP\text{-}GAL4\}LL7/ TM6b,GFP$), *act5C-GAL4* (stock 4414, $y^1 w^*$; $P\{w^{+mC} = Act5C\text{-}GAL4\}25FO1/CyO, y^+$;), *nub-GAL4* (stock 25754; $P\{UAS\text{-}Dcr\text{-}2.D\}1, w^{1118}$; $P\{GawB\}nub\text{-}AC\text{-}62$), *69B-GAL4* (stock 1774; $w^*$;; $P\{GawB\}69B$), *UAS-idgf1^{RNAi}* (stock 57508, $y^1 sc^* v^1$; $P\{TRiP.HMC04823\}attP40$), *UAS-idgf2^{RNAi}* (stock 55935, $y^1 sc^* v^1$; $P\{TRiP.HMC04223\}attP40$), *UAS-idgf3^{RNAi}* (stock 67226, $y^1 sc^* v^1$; $P\{TRiP.HMC06327\}attP40$), *Df(3L)Exel6084* (stock 7563, $w^{1118}$;; $Df(3L)Exel6084$, $P\{XP\text{-}U\}Exel6084/TM6B$). Stocks obtained from the Vienna *Drosophila* Resource Center were: *UAS-dis3L2^{RNAi}* (stock v51854, $w^{1118}$; $P\{GD9240\}v51854$; and stock v100322,;; $P\{KK105902\} VIE\text{-}260B$) and *Dis3L2^{CD}* (stock 312503). *en-GAL4* (A kind gift from Paul Martin,; *engrailed-GAL4,UAS-GFPactin/Cyo;*). To overexpress Dis3L2 the stock $w^*$;; $P\{GSV3\}GS6090/TM6$ was purchased from the Kyoto Stock Center (stock 200902), however, a mutation outside the Df6084 locus caused homozygous lethality; this was repaired by recombination.

### Generation of CRISPR mutants

A gRNA to the first exon of *dis3L2* was cloned into the Bbs1 site of pCFD3-dU6:3. The resulting vector was then sent to BestGene Inc for injection into *vas-Cas9* embryos (BDSC 51323 $y^1 M\{vas\text{-}Cas9\}ZH2A w^{1118}$). Potential mutant chromosomes were balanced and subject to PCR

screening using the gRNA as a forward primer (see S1 Table for all primers/gRNA used in this study). Potential mutant stocks were sequenced at Eurofins Genomics. An 8bp frame shift mutation was identified and its lack of Dis3L2 confirmed by Western blotting. This line was named $dis3L2^{12}$.

## Western blotting and generation of anti-dDis3L2 antibody

A polyclonal antibody was raised against the N-terminal tagged Dis3L2 peptide consisting of the first 198 amino acids of the *Drosophila* protein. pDNR plasmid-B527503, containing the cDNA for *Drosophila* CG16940 RC gene (Isoform PC), was used as template for the amplification of the *Drosophila dis3L2* gene in the construction of two N-terminal fusion proteins with different tags (GST-Dis3L2 and MBP-Dis3L2), that were expressed and purified in *E. coli*. The GST- and MBP-tagged proteins were used for the immunization in the goat and for the affinity purification of the anti-serum, respectively (by SICGEN, Portugal).

Western blots were performed in samples of adult females (x3), 120hr L3 larvae (x5) or wing imaginal discs (x60). Tubulin was used as a loading control on all blots. All blots apart from those assessing AKT T342 phosphorylation were blocked in Odyssey Blocking Buffer (PBS) (LI-COR cat. no. 927–40000) with antibody incubations performed in Odyssey Blocking Buffer containing 0.1% Tween. Mouse anti-Tubulin primary antibody (Sigma, cat. no. T9026) was used at 1:2000 dilution. Goat anti-Dis3L2 (produced in this study) was used at 1:2500. Anti-mouse, anti-rabbit and anti-goat fluorescent secondary antibodies were used at 1:20,000 (LI-COR Donkey anti-mouse IRDye 800CW (cat. No. 925–32212), Goat anti-rabbit IRDye 680RD (cat. no. 926–68071) and Donkey anti-goat IRDye 680RD (cat. no. 925–68074)). Anti-human DIS3L2 (hDIS3L2) was used at 1:500 (Novus Biologicals cat. no. NBP-1-84740). Anti-GAPDH (Abcam cat. no. ab8245) was also used at 1:10,000 as a human specific loading control along with the anti-mouse IRDye 800CW secondary antibody above. Blots assessing AKT T342 phosphorylation were washed in 0.1% TBS-Tween and blocking and antibody incubation steps performed in 5% BSA in 0.1% TBS-Tween. Anti-pAKT (T342) (Phosphosolutions cat. no. p104-342) was used at 1:500. Detection and quantification were performed using the LI-COR Odyssey FC imager and Image Studio (version 5.2) respectively. Uncropped blots are shown in S2 File.

## Phospho-specific western blotting in HEK-293T cells

Blots were performed on cell pellets from the specified time post transfection or treatment. Samples were ran on 4–12% Novex gradient gels (Invitrogen cat. no NP0321BOX) or Mini-Protean TGX Stain-Free 4–15% gels (Bio-Rad cat. no. 4568084). All membranes were blocked in 3% BSA in TBS with antibody incubation in 3% BSA in 0.1% TBS-Tween. Primary antibodies used were as follows: Anti-pAKT (T308) (Cell Signalling cat. no. 1303S), anti-pAKT (S473) (Cell Signalling cat. no. 4060S), anti-AKT (Cell Signalling cat. no. 4685S) anti-pp42/44 MAPK (T202/Y204) (Cell Signalling cat. no. 9101S), anti-pPRAS40 (T246) (Cell Signalling cat. no. 2997S), anti-PRAS40 (Cell Signalling cat. no. 9644S), pS6-Ribosomal Protein (Cell Signalling cat. no. 2215S), anti-p4E-BP (T37/46) (Cell Signalling cat. no 2855S), anti-p4E-BP (S65) (Cell Signalling cat. no. 9451S), anti-p4E-BP (T70) (Cell Signalling cat. no. 5078S) and anti-4E-BP (Cell Signalling cat. no. 2610S) were all used at 1:1000. Mouse anti-Tubulin primary antibody (Sigma, cat. no. T9026) was used at 1:2000 dilution. Goat anti-Rabbit IRDye 680RD (LI-COR cat. No. 925–68071) and Donkey anti-mouse IRDye 800CW (cat. No. 925–32212) were used at 1:20,000. Detection and quantification were performed using the LI-COR Odyssey FC imager and Image Studio (version 5.2) respectively. Differential phosphorylation status was assessed by normalising signal in DIS3L2$^{KD}$ cells to the respective scrambled control ran on the same gel. Uncropped blots are shown in S2 File.

## Assessing lifespan

1 day old *dis3L2^{wt}* and *dis3L2^{12}* flies were transferred into new food. Every 3–4 days flies were transferred into new food and the number of deceased flies were counted. This continued until all flies had deceased.

## Measurement of wing and wing imaginal disc area

Adult flies were aged to between 1 and 2 days old for all wing measurement experiments. A single wing was cut from each fly and stored in isopropanol for 1 hour before being mounted in DPX (Sigma cat. no. 06522). All results shown are for male wings, however, female wings also showed the same phenotypes. Mounted wings were measured using Axiovision 4.7 on an Axioplan microscope (Carl Zeiss). Imaginal discs were dissected in PBS from 120hr old L3 larvae, aged using a 1 hour egg lay. Dissected discs were mounted in 85% Glycerol on Poly-L-Lysine slides. Imaginal disc area was measured using Axiovision 4.7 on an Axioplan microscope (Carl Zeiss). Salivary glands were prepared and imaged using the same methodology as used for imaginal discs.

## Measurement of whole flies and larvae

Control and mutant female flies were left for 1 hour to lay on grape agar plates at 25°C. 24 hours later 20 L1 larvae were transferred into food vials and left to develop in uncrowded conditions. 5 vials (100 larvae total) were set up for each genotype. Eclosing adults were then aged to 1 day old, individually weighed and photographed. Specific, defined regions of each male fly were then measured using ImageJ (S7 Fig). Larval mass was calculated by washing 120hr L3 larvae (cultured as above) in PBS then weighing individually in an Eppendorf. Larval footprint was calculated from the weighed flies following 30 min incubation in 100% Methanol where the larvae elongate. Larvae were then photographed and their "footprint" was measured in pixels using ImageJ.

## SUnSET labelling

Surface sensing of translation (SUnSET) was performed on *dis3L2^{wt}* and *dis3L2^{12}* wing imaginal discs. Wing discs were dissected in batches of 30 and incubated in fully supplemented Shields and Sang M3 insect medium (Sigma-Aldrich, cat. no S8398) containing 2μg/ml Puromycin (Sigma-Aldrich, cat. no. P8833) for 1 hour at 25°C with rotation. Western blotting was performed to determine the levels of Puromycin incorporation with Tubulin as a loading control. Mouse anti-Tubulin primary antibody (Sigma-Aldrich, cat. no. T9026) was used at 1:2000 dilution. Mouse anti-Puromycin (clone 12D10) primary antibody (Merck Millipore, cat. no. MABE343) was used at 1:1000 dilution. Anti-mouse IRDye 800CW secondary antibody (LI-COR Biosciences, cat. no 926–32210) was used at 1:20,000 dilution to detect both primary antibodies. Each sample was run in parallel on two gels/membranes so the Tubulin band could be distinguished from Puromycin containing peptides. Quantification of Tubulin (46kDa) and Puromycin peptides (from smallest size visible to 240kDa) was achieved using LI-COR Biosciences Image Studio software. In each case Puromycin signal in *dis3L2^{12}* tissues was compared to *dis3L2^{wt}* samples ran on the same gel. Uncropped blot is shown in S2 File.

## Fertility assays

Individual virgin, 3 day old male flies of the test genotype were crossed to 2 virgin *dis3L2^{wt}* females. Flies were left for 48 hours to mate, and then removed. Eclosing progeny were

counted for 7 days. In all crosses using *dis3L2*[12] homozygous and hemizygous male parents, many eggs were laid by the *dis3L2*[wt] females but none were fertilised.

## Generation of *UAS-dis3L2* and *UAS-idgf2* lines

*Drosophila dis3L2* was amplified from pDNR-DUAL (DGRC clone BS27503) using two primer pairs; (1) PC which amplified the full coding region of *dis3L2* capable of encoding both isoforms and (2) PA which only amplified the coding region of the shorter isoform (PA). These were cloned into the AgeI and XbaI sites of pUASP-attB. Constructs were then sent to Best-Gene Inc for injection into a recipient line with an attP site at 51C1 (BDSC 24482; M{3xP3-RFP.attP'}ZHC51C). Lines where the insert was successfully integrated were confirmed by PCR and balanced. To produce the *UAS-dis3L2*[ND] line site-directed mutagenesis was performed using the Agilent Quickchange Lightning mutagenesis kit (Agilent, cat. No. 210518), and primers containing the G1738A mutation (D580N) on the pUASP-attB vector containing the PC insert. Human *DIS3L2* (Ensembl isoform 202) was amplified from oligo(dT) primed cDNA produced from hFOB cells (ATCC CRL-11372) and cloned into the NotI and SpeI sites of pUASP-attB. Injection and subsequent processing were performed as above. *UAS-idgf2* was produced by amplifying *idgf2* (CDS and 3'UTR) from cDNA from *dis3L2*[wt] wing imaginal discs and subsequent cloning to the NotI and SpeI sites of pUASP-attB. The created vector was then injected into attP40 (y[1] w[67c23]; P{CaryP}attp40) and balanced. The sequences of all primer sets used are described in S1 Table.

## Mitotic index

Immunocytochemistry was performed essentially as described in [1] on 120hr L3 wing imaginal discs aged using 1 hour egg lay periods. Images were taken with a Leica SP8 confocal microscope. Anti-Phosphohistone H3 (Cell Signalling, cat no. 9701) was used at 1:300 dilution. Cy3-conjugated monoclonal Donkey anti-mouse IgG secondary antibody was used at 1:400 (Jackson ImmunoResearch, cat. no.715-165-151). The number of nuclei undergoing mitosis were counted using the ImageJ plugin DeadEasy MitoGlia [42]. The mitotic index was then calculated for each disc by dividing the number of cells in M phase by the area of the disc.

## Wortmannin assay in *Drosophila*

Fresh food containing Formula 4–24 Instant *Drosophila* Medium (Carolina cat no. 173200), yeast and bromophenol blue was made each day with either the desired concentration of Wortmannin (5μM or 10μM) or an equal volume of DMSO. 4 hour egg lays were performed on grape agar plates and an equal number of larvae were transferred to Wortmannin or control food for each genotype at specific developmental time points (24hr, 48hr and 72hr after egg lay) then left to develop at 25˚C. Consumption of the drug was confirmed by visualising the blue food in the gut. Wings were then cut and measured from eclosing adults as outlined above.

## RNA-seq sample preparation and analysis

RNA was extracted from 60 wing imaginal discs dissected from 120hr L3 larvae. Three replicates of control and mutant were dissected. In these experiments, control samples were collected from a stock containing a 6bp mutation in *dis3L2* that does not affect wing size or induce any other obvious phenotype (S4A Fig). Downstream qRT-PCR validation samples were collected from the *dis3L2*[wt] stock which contains the wild-type sequence and were grouped together with the sequenced samples for downstream analysis (*dis3L2*[wt]). To ensure

accurate aging, 3 hour egg lays were used and wing discs were dissected in Ringers solution 120hrs after egg laying. RNA was extracted using a miRNeasy RNA extraction kit (Qiagen, cat. no. 217084), with on-column DNase digestion (Qiagen, cat. no. 79254). RNA concentrations were measured on a NanoDrop One spectrophotometer (Thermo Scientific). RNA integrity was assessed on an Agilent 2100 Bioanalyser.

400ng total RNA was depleted for ribosomal RNA as performed in [43] with the resulting RNA sent to Leeds Genomics for library preparation using the Illumina TruSeq Stranded protocol. Subsequent libraries were run in a paired-end sequencing run on a HiSeq 3000 generating between 36 and 44 million reads per sample. Raw RNA-sequencing files have been deposited in ArrayExpress. Accession number: E-MTAB-7451

Sequencing quality was assessed using FastQC c0.11.7 (http://www.bioinformatics. babraham.ac.uk/projects/fastqc/) and adapters were removed using Scythe v0.993b (https:// github.com/vsbuffalo/scythe). Further quality control and read trimming was achieved using Sickle v1.29 (https://github.com/najoshi/sickle). The remaining high quality reads were mapped to the *Drosophila melanogaster* genome from Flybase (r6.18 [44]) using HiSat2 v2.1.0 [45]. Differential expression was completed and normalised FKPM values were generated using the Cufflinks pipeline [46]. Individual replicates from each condition showed good consistency (S1 File). Due to issues with statistical outputs from these pipelines we used individual replicate comparisons to minimise false positives when specifying that a gene is differentially expressed. All mutant replicates required a fold change of >1.34 compared to all control replicates. A fold change cut off of >1.34 was selected as this was the smallest fold change deemed significant from the Cuffdiff output. Alignment results and non-default parameters used during the analysis are shown in S1 File. Raw sequencing files from [11] were processed in the same analysis pipeline to allow direct comparison.

## RNA extraction and qRT-PCR

RNA extractions were performed using a miRNeasy RNA extraction kit (Qiagen, cat. no. 217084), with on-column DNase digestion (Qiagen, cat. no. 79254). RNA concentrations were measured on a NanoDrop One spectrophotometer (Thermo Scientific).

For qRT-PCR, 1μg of total RNA was converted to cDNA in duplicate using a High Capacity cDNA Reverse Transcription Kit (Life Technologies, cat. no. 4368814) with random primers. A control "no RT" reaction was performed in parallel to confirm that all genomic DNA had been degraded. qRT-PCR was performed on each cDNA replicate in duplicate (i.e. 4 technical replicates in all), using TaqMan Universal PCR Master Mix, No AmpErase UNG (Life Technologies, cat. no. 4324018) and an appropriate TaqMan assay (Life Technologies). For custom pre-mRNA assays, the pre-mRNA sequence of the desired target area was submitted to Life Technologies' web-based Custom TaqMan Assay Design Tool as in [47] (S1 Table). *rpL32* (*rp49*) was used for normalisation.

## Human cell culture and growth curve analysis

HEK-293T and U-2 OS cells were cultured in Dulbecco's Modified Eagle's Medium/F12 (DMEM/F12 –Gibco cat. no. 21331–020) supplemented with 10% foetal bovine serum (PAN Biotech, cat. no. P40-37500), 2mM L-Glutamine (Gibco cat. no. 25030–024) and antibiotics (100IU/mL penicillin, 100μg/mL streptomycin, Sigma Aldrich cat. no. 15140–122), at 37˚C in a 5% $CO_2$ humidified incubator. For growth analysis of DIS3L2 knockdown, $3\times10^5$ cells were plated in 6 well plates and transfected with 30pmol siRNA (DIS3L2 (Invitrogen cat. no. AM16708) or scrambled control (Invitrogen cat. no. AM4611)) using Lipofectamine RNAiMAX reagent (ThermoFisher cat. no. 13778030). Cell number was counted in triplicate every 24hrs for

144hrs. For Wortmannin treatment analysis cells $3 \times 10^5$ HEK-293T cells were plated and treated with siRNA as above. 250nM Wortmannin or an equal volume of DMSO was then added 20, 44, and 68hrs, post-transfection. Cells were counted in triplicate every 24hrs following transfection.

## Statistical tests

All statistical analyses were performed in either R v3.5.1 or GraphPad Prism 7. Two-sided two-sample t-tests were used to compare the means of single test groups to single control groups. If multiple comparisons were required, a one-way ANOVA was performed with a post-test to compare the means of each possible pair of samples.

## Supporting information

**S1 Fig. Generation and characterisation of a CRISPR-Cas9 *dis3L2* null mutant. A)** DNA sequencing of the control *dis3L2*$^{wt}$ and mutant *dis3L2*$^{12}$ lines where the arrow denotes the site of mutation. **B/B'/B")** The mass (B), footprint (B') or salivary gland area (B") of *dis3L2*$^{12}$ 120hr old L3 larvae are not significantly different from *dis3L2*$^{wt}$ controls. n = 16–31, error bars represent 95% CI, ns = p>0.05. **C)** Male flies lacking Dis3L2 are infertile. In all crosses no progeny was observed for those using homozygous (*dis3L2*$^{12}$), hemizygous (*dis3L2*$^{12}$/*Df*) or *dis3L2* knockdown (*Tub>dis3L2*$^{RNAi}$) males. Error bars represent 95% CI, ns = p>0.05. For all crosses males of each genotype were crossed to virgin *dis3L2*$^{wt}$ females. **D/D')** *dis3L2*$^{12}$ mutant males **(D)** and females **(D')** have a reduced lifespan compared to *dis3L2*$^{wt}$ isogenic controls. n = 47–57, error bars represent 95% CI with significant differences demonstrated by the lack of overlapping error bars.
(TIF)

**S2 Fig. Ectopic expression of Dis3L2 rescues *dis3L2*$^{12}$ induced overgrowth. A)** Quantification of Western blots performed on 3, 1-day old, female flies of the demonstrated genotypes. n = 4–10, error bars represent SEM. **B)** Whilst specific re-expression of Dis3L2 in the wing (with *nub-GAL4*) rescues wing overgrowth it does not rescue organism overgrowth. n = 10–44, error bars represent 95% CI, ** = p<0.01, **** = p<0.0001. **C)** Re-expression of wild-type Dis3L2 (PC) but not catalytically dead Dis3L2 (ND) in the posterior compartment of the wing/ wing imaginal disc results in a specific rescue of the posterior area of the wing whilst the anterior area remains significantly larger. n = 24–44, error bars represent 95% CI, ns = p>0.05, *** = p = <0.001, **** = p = <0.0001. **D/E)** Mild overexpression of Dis3L2 using the rescue *UAS--Dis3L2*$^{PA/PC}$ constructs does not have a major effect on wing area when driven by *nub-GAL4* **(D)** or *en-GAL4* **(E)**. n = 16–47, error bars represent 95% CI. **F)** Human DIS3L2 is successfully expressed in *Drosophila* from the *UAS-hDIS3L2* construct. Two human osteosarcoma cells lines were used as positive controls (U-2 OS and SAOS-2). Protein lysate was prepared from $1 \times 10^6$ cells or 4, 1-day old, adult females. hDIS3L2 is observed specifically in the human cells and female flies where *UAS-hDIS3L2* had been driven by *Tub-GAL4* at 25°C (*Tub>UAS-hDIS3L2*). No product was observed in the parental controls (*UAS-hDIS3L2* and *Tub-GAL4/ TM6*), confirming no 'leaky' expression from the *UAS-hDIS3L2* line. Tubulin used as a loading control for all samples as the antibody detects both human and *Drosophila* protein. hGAPDH used as an additional loading control specifically for the human cell line samples.
(TIF)

**S3 Fig. Summary and highlights of the analysis of the RNA-seq experiment. A)** Summary of the number of transcripts showing up- and downregulation in *dis3L2*$^{12}$ wing imaginal discs. A fold change cut off of >1.34 was selected as this was the smallest change deemed significant by Cuffdiff. Inf change represents transcripts that were only detected in a single condition. **B)**

Integrative Genomics Viewer screenshot showing accumulation of unprocessed *RNaseMRP: RNA* transcripts in *dis3L2^12* tissues. **C)** Scatter plot of misregulated genes coloured by significant gene ontology categories. "None" represents genes that belonged to a category that was not significantly enriched. **D)** Strip plots showing replicate FPKM values for each of the *idgf* family in *dis3L2^wt* and *dis3L2^12* wing discs. Only *idgf1*, *idgf2* and *idgf3* show changes in expression. (TIF)

**S4 Fig. Additional information from RNA-seq data. A)** The 6nt deletion in the control stock used for RNA-sequencing does not affect Dis3L2 protein expression or wing area in male or female flies. n = 19–33, error bars represent 95% CI, ns = p<0.05. **B)** Transcripts misexpressed in *dis3L2^12* wing imaginal discs have significantly shorter 3' UTRs (446nt vs 598nt, Welch 2 sample t-test p = 2.078e-05). 5' UTRs (337nt vs 365nt) and the coding sequence (CDS, 1712nt vs 1837nt) show no difference in length between transcripts. Median and upper and lower quartile are represented by horizontal lines with maximum and minimum values shown vertically. **C)** All validated mRNAs show significant increases in expression in *dis3L2^12* hemizygote wing imaginal discs compared to *dis3L2^wt* wing discs. n = 3–6, error bars represent SEM, p<0.05 for all. **D)** 10% of transcripts misexpressed in *dis3L2^12* wing discs are also misexpressed in *dis3L2* mutant testes [11]. **E)** MEME analysis identifies U-rich and CA-rich motifs are significantly enriched in likely Dis3L2 targets. U-rich: E-value = $5.2e^{-6}$, found in 40.6% of submitted sequences. CA-rich: E-value = 0.0012, found in 14.1% of submitted sequences. **F)** A G-rich motif present in 23.1% of control sequences (sequences that show no change and do not coprecipitate with Dis3L2 [13]) is absent in Dis3L2 target sequences. E-value = 0.016. (TIF)

**S5 Fig. Assessing *idgf2* levels in knockdown, mutant and rescue tissues. A)** Ubiquitous knockdown of *idgf1*, *idgf2* and *idgf3* by driving specific *UAS-RNAi* constructs with *Tub-GAL4* results in >90% knockdown for all targets. n = 3, p<0.0004 for all, error bars represent SEM. **B)** Re-expression of Dis3L2 in *dis3L2^12* wing imaginal discs (*nub>PC; dis3L2^12*) results in a reduction of *idgf2* mRNA to a level not significantly different from *dis3L2^wt* tissues. n = 5–6, error bars represent SEM, \*\*\* = p<0.001, ns = p = 0.4748. **C)** *idgf2* mRNA is significantly increased in expression in the wing imaginal discs of an independent line carrying a CRISPR generated catalytic dead mutation in the endogenous *dis3L2* locus (*dis3L2^CD*)[13]. n = 4–6, error bars represent SEM, \*\*\*\*p<0.0001, \*\*\*p = 0.0006. **D)** Driving *UAS-Idgf2* with *69B-GAL4* results in a significant increase in *idgf2* mRNA in the wing imaginal disc (*UAS-idgf2/+; 69B-GAL4/+*) compared to controls. Controls include both parental lines. n = 6, 95% CI, \*\*\*\* = p<0.0001. (TIF)

**S6 Fig. Knockdown of DIS3L2 and validation of Wortmannin activity in human cells. A)** Knockdown of DIS3L2 in human kidney HEK-293T cells is observed 48 hours after transfection and is retained until at least 144hrs post-transfection. Maximal knockdown is observed 72 hours post transfection. n = 4, error bars represent SEM, \*\*\*\* = p<0.0001, \*\* = p<0.01. **B)** Knockdown of DIS3L2 in human osteosarcoma U-2 OS cells is observed 48 hours after transfection and is retained until at least 144hrs post-transfection. Maximal knockdown is observed 72 hours post transfection. n = 4, error bars represent SEM, \*\*\*\* = p<0.0001. **C)** Representative image and quantification of Western blots assessing total protein levels of AKT, PRAS40 and 4E-BP in DIS3L2 knockdown or scrambled control HEK-293T cells 72hrs post transfection. n = 7, error bars represent SEM, \* = p = 0.00379. Also shown is phosphorylated protein levels normalised to total protein levels. n = 7, error bars represent SEM, \* = p = 0.0214. PRAS40 and 4E-BP also show a trend towards more phosphorylated protein but this is not statistically significant (p>0.05). **D)** Wortmannin treatment reduces pAKT (T308) pPRAS40 and

p4E-BP (T37/26) signal in DIS3L2 Knockdown HEK-293T cells 48hrs post transfection. Representative images and quantification of Western blots in $DIS3L2^{KD}$ or Scrambled control HEK-293T cells treated with either DMSO or 250nM Wortmannin. n = 4, error bars represent SEM, * = p<0.05, ** = p<0.01, *** = p<0.001.
(TIF)

**S7 Fig. Regions measured to assess male fly size.** Measurements taken between the arrows using ImageJ.
(TIF)

**S1 Table. Primers used in this study.**
(DOCX)

**S1 File. Additional RNA-seq information.**
(DOCX)

**S2 File. Uncropped western blots.**
(DOCX)

## Acknowledgments

The authors would like to thank Chris Jones, Oliver Rogoyski, Elisa Bernard, Jose Pueyo-Marques and Helen Stewart for helpful discussions plus critical reading of the manuscript. We would also like the thank Clare Rizzo-Singh for technical help.

## Author Contributions

**Conceptualization:** Benjamin P. Towler, Sarah F. Newbury.

**Data curation:** Benjamin P. Towler.

**Formal analysis:** Benjamin P. Towler, Amy L. Pashler.

**Funding acquisition:** Simon J. Morley, Cecilia M. Arraiano, Sarah F. Newbury.

**Investigation:** Benjamin P. Towler, Amy L. Pashler, Hope J. Haime, Katarzyna M. Przybyl, Sandra C. Viegas, Rute G. Matos.

**Methodology:** Benjamin P. Towler, Amy L. Pashler, Sarah F. Newbury.

**Project administration:** Sarah F. Newbury.

**Resources:** Simon J. Morley, Cecilia M. Arraiano, Sarah F. Newbury.

**Software:** Benjamin P. Towler.

**Supervision:** Benjamin P. Towler, Simon J. Morley, Cecilia M. Arraiano, Sarah F. Newbury.

**Visualization:** Benjamin P. Towler, Sarah F. Newbury.

**Writing – original draft:** Benjamin P. Towler, Sarah F. Newbury.

**Writing – review & editing:** Benjamin P. Towler, Sandra C. Viegas, Simon J. Morley, Sarah F. Newbury.

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
