## [Decision Letter · Decision Letter 0]

27 Mar 2020

Dear Dr Newbury,

Thank you very much for submitting your Research Article entitled 'Dis3L2 regulates cellular proliferation through a PI3-Kinase dependent signalling pathway' to PLOS Genetics. Your manuscript was fully evaluated at the editorial level and by independent peer reviewers. The reviewers appreciated the attention to an important problem, but raised some substantial concerns about the current manuscript. Based on the reviews, we will not be able to accept this version of the manuscript, but we would be willing to review again a much-revised version. We cannot, of course, promise publication at that time.

If you decide to revise the manuscript for further consideration at PLOS Genetics, please aim to resubmit within the next 60 days, unless it will take extra time to address the concerns of the reviewers, in which case we would appreciate an expected resubmission date by email to plosgenetics@plos.org.

[LINK]

We are sorry that we cannot be more positive about your manuscript at this stage. Please do not hesitate to contact us if you have any concerns or questions.

Yours sincerely,

Ville Hietakangas

Associate Editor

PLOS Genetics

Gregory Barsh

Editor-in-Chief

PLOS Genetics

Reviewer's Responses to Questions

**Comments to the Authors:**

Reviewer #1: In this paper Towler et al show that fly CRISPR mutants for the exoribonuclease Dis3L2 show overgrowth phenotypes in imaginal tissues and final adult organ size. Similarly, they find that knockdown of Dis3L2 in HEK293 cells also causes an increase in proliferation. Using a combination of RNA seq and analyses of published

Overall I thought that this was a good paper, showing the characterization of a new growth regulator. Most of my comments should be easy to address

1. The authors use a nice combination of cell culture and Drosophila experiments to ask how loss of Dis3L2 promotes overgrowth. The pinpoint upregulation of Idgf2 and upregulation of PI3K signaling as mechanisms. But it's not entirely clear whether these two effects are related. I think a couple more analyses could be done:

a) Does loss of Dis3L2 lead to upregulation of the homolog of Idgf2 in the HEK293 cells? If not, then it would suggest that the Idgf2 and PI3K/Akt mechanisms are independent.

b) Does loss of Dis3L2/overexpression of Idgf2 in Drosophila lead to increased Pi3K/Akt activity? This can be tested by looking at phospho-Akt western blots on discs or by using qRT-PCR to look at FOXO target genes in discs (these would be decreased with increased PI3k/Akt signaling). The wortmanin data indicate that Pi3K is required for Dis3L2/idgf2 regulated growth, but this may not mean that either or both actually regulate the pathway. If Pi3k/Akt activity isnt increased then it woudl again suggest idgf2 and Pi3k work in parallel, independent pathways to control growth.

2. Other than the imaginal tissues, are any other larval tissues increases in growth (e.g. fat body, salivary gland) or is overall body growth/development increased. These are all sensitive to PI3K signaling, but it could be that Dis3L2 expression and function is limited to specific tissues, which would be interesting.

Reviewer #2: In this MS, Towler and cols study the role of Dis3L2 in growth control. They use the imaginal tissues of Drosophila as a primary model to determine the mechanisms used by this exoribonucease in the regulation of tissue size. The authors identify Idgf2 and PI3K signaling as elements mediating the growth regulatory role of Dis3L2. Growth control is a central aspect in biology and therefore this study is potentially interesting. However, the study is very preliminary and is not suitable for publication in Plos Genetics in its current form. Some issues need to be addressed.

Specific comments are:

It would be helpful showing in Fig 1 schematic representation of the dis3L2 gene indicating the position and nature of the dis3L2-12 mutation.

The authors should introduce the SuNSET labeling approach in the text. This would facilitate the understanding of this experiment.

Line 147, the authors mention that act-Gal4 allowed expression of Dis3L2 to levels consistent with that of the control line. What do the authors mean with this? In the Fig is very clear that the levels are reduced, as compared to the control, which I guess is the +- line. The authors should clarify this.

What is the phenotype of over-expressing dis3L2 in the wing? It should be shown in Fig 2.

What are the protein levels obtained when expressing the line that lacks catalytic activity? This should be shown In the western shown in Fig 2a, as well as the levels obtained when expressing the human protein. Otherwise, the results obtained are difficult to interpret and it is difficult to compare between the different conditions analyzed.

The authors suggest that different rescue obtained at different temperatures could be due to the human protein being more stable at 37C. It is worth noting that, generally, Gal4 driver are stronger at 29C than at 25C. I would suggest to consider and mention that in the text as a potential explanation.

The authors use the anterior compartment as an internal control in the overexpression experiments showed in Fig 3. However, compartments accommodate in some cases to defects in growth (see PMID, 21179433). To prevent a misinterpretation due to potential final size effects mediated by accommodation, and to be consistent with previous Figs, would suggest to also use nub-Gal4 to perform this analysis. This comment also applies to Fig 5.

As in Fig 1, representative images of adult wings should be provided in the Figure.

Why then the authors switch to a different Gal4 driver (69B) to study PH3? The authors should stick to the use of the same genetic tools to make the results obtained comparable. Using PH3 as a proxy of cell proliferation is not sufficient to conclude that cells proliferate less. PH3 labels cells in mitosis and, if mitosis occurs faster, the number of PH3 positive cells will be reduced with a similar final mitotic rate. Besides, discs of different ages have different proliferation rates and slight deviation in the timing of those discs can have a strong impact in the result obtained. The authors should use alternative tools as EdU incorporation as well as induce clones and calculate the cell doubling time (see PMID, 9657151) in the different conditions analyzed to present these results in a convincing manner. These analysis should also be performed in the loss of function of Dis3L2 and in a context of Idgf2 over-expressio.

The authors should validate by qPCR the top 30 genes shown in Fig 4. This would provide information about the sensitivity of the approach.

To link dis3L2 with PI3-K in the control of cell proliferation in the wing disc (Fig 7), the authors should analyze whether the manipulation of dis3L2 and Idgf2 (GOF and LOF experiments) results in changes in PI3K activity.

Reviewer #3: Review of Towler et al. “Dis3L2 regulates cellular proliferation through a PI3-K dependent signaling pathway”

In this manuscript, Towler et al. show that loss of the conserved 3’-5’ exoribonuclease Dis3L2 leads to increased cellular proliferation and overgrowth in Drosophila wing imaginal discs and HEK-293T cells. The authors present evidence that increased proliferation is due in part to PI3K activation, as wortmannin treatment reduces organ size and mitotic index in Dis3L2 mutant fly wings and cell number in HEK-293T cells with Dis3L2 knockdown. The investigators identified transcripts affected by loss of Dis3L2 by RNA sequencing of control and Dis3L2 mutant wing discs. This led to the identification of imaginal disc growth factors – Idgf1, Idgf2 and Idgf3 – as possible targets of the exoribonuclease activity of Dis3L2. Epistasis experiments to knockdown each Idgf individually in Dis3L2 mutants, generated by the authors using CRISPR/Cas9, show that Idgf2 is an important positive regulator of growth in wing discs lacking Dis3L2.

Overall, this study illustrates the importance of RNA processing as a contributor to cell and organ growth. The study would be strengthened by the addition of additional experiments that would shore up the authors’ findings and connect Idgf2 with Dis3L2 and PI3K activation.

Major points.

Tissue growth.

1. Tissue- or compartment-specific manipulations of Dis3L2 or Idgf2 alter growth in a tissue- autonomous manner, as shown in Figures 2, 3 and 5. Do the authors find tissue-nonautonomous growth regulation by Dis3L2? For example, are leg lengths still increased in whole-animal Dis3L2 mutants that have wing-specific re-expression of wild type Dis3L2? This should be examined because a secreted factor – Idgf2 – accounts for growth phenotypes in Dis3L2 mutants. Similarly, does overexpression of Idgf2 using 69B-GAL4 induce growth only in wings?

2. Changes in wing cell number in response to Dis3L2 and Idgf2 manipulations are inferred by assessing mitotic index (Figures 3D and 5E). The authors should determine whether these manipulations also alter cell size, as might be expected if PI3K is activated. Direct measures of cell number in wild type and Dis3L2 adult wings should be provided.

RNA-sequencing.

3. The authors used flies with a 6-nt deletion of Dis3L2 as a control for their RNA-sequencing. Data should be provided in a supplement showing that this allele is wild type with regard to animal size, wing growth, and Dis3L2 protein expression.

4. RNA sequencing data suggest that Dis3L2 regulates transcripts with a shorter than average 3’-UTR. The authors find enrichment of a U-rich motif in the 3’-UTR of Dis3L2-regulated transcripts but absence of a G-rich motif. Are any of these features found in the Idgf2 3’-UTR?

Dis3L2 and PI3K/mTOR signaling.

5. The connection between Dis3L2, Idgf2, and PI3K activity is inferred from results showing that wortmannin feeding partially rescues wing size in Dis3L2 mutants and in flies overexpressing Idg2 in wing. The authors should use immunocytochemistry for tGPH, a PIP3 reporter, or for phosphorylated Akt to determine whether insulin signaling is stimulated by compartment-specific Dis3L2 knockdown or Idgf2 overexpression. Wortmannin can also inhibit mTORC1, so it is particularly important to see whether the upstream part of the PI3K-mTORC1 pathway is induced by loss of Dis3L2 and inhbited by wortmannin feeding.

6. Signaling experiments in HEK-293T cells are uninterpretable without additional controls.

First, total levels of the phosphorylated kinases and substrates measured in Figure 6 and Supplemental Figure 6 must be shown. This is especially important because the authors describe that the PI3K/Akt pathway is significantly differentially expressed in HEK-293T cells lacking Dis3L2 (although direction of change is not specified). It is possible that an increase in basal levels is driving the pathway, rather than autocrine signaling.

Second, data in Figure 6F showing rescue of elevated phosphorylation of Akt and TORC1 targets in wortmannin-treated Dis3L2 knockdown HEK-293T cells are impossible to interpret without vehicle controls on the same blot. The experiment should be repeated. Vehicle controls are also needed for the 48h and 72h time points shown in Supplemental Figure 6C.

Third, phospho-Thr308 Akt should be sensitive to wortmannin treatment as well. This should be assessed.

Data presentation.

7. Data should be presented as absolute measurements where possible. See, for example, Figure 1 where growth measurements are presented as percent increases over wild type rather than mm2 for disc or wing area or mg body weight. The percent increase can be described in the text, as it is currently.

8. Exact sample sizes (or a range) for each set of data should be reported in Figure Legends, rather than “n ≥ 22”, for example.

Discussion points.

9. The results from immortalized cell lines should be interpreted more carefully. HEK-293T cells are not a physiological model of kidney, and differences between HEK-293T cells and U2OS cells may not reflect tissue of origin, but genetic differences acquired as a result of immortalization or tumorigenesis.

10. Pre-mRNA levels of Idgf2 are increased by Dis3L2. As Dis3L2 is thought to act post-transcriptionally, this suggests a secondary effect caused by loss of Dis3L2. Furthermore, wortmannin treatment of Dis3L2 mutants partially rescues wing size without altering Idgf2 mRNA levels. The authors should comment on these discrepancies in the Discussion.

11. The authors discuss the two activating phosphorylation sites on Akt and their relative contributions to Akt activity (lines 458-467). Indeed, a number of studies show that phosphorylation of the activation loop threonine (308 in mice/342 in flies) is critical for Akt activity, and that phosphorylation of the hydrophobic motif serine (473 in mice/505 in flies) is important for full activity and control of a subset of substrates. These papers should be cited. See: Cheng et al., JCB, 2011; Guertin et al., Dev Cell, 2006; Hietakangas and Cohen, G&D, 2007; Jacinto et al., Cell, 2006; Kumar et al., Diabetes 2010; Lee and Chung, BBRC, 2007.

Minor points.

1. Careful proofreading of the manuscript is necessary. A few examples of errors to correct are:

Defining abbreviations (line 77), grammar (lines 182-184, 406, 436-440), subject/verb agreement (lines 189, 272-273, 494), spelling (line 349)

2. Molecular weights should be indicated on Western blots.

3. Scale bars should be added to Figures 1A, 3C’, and S7.

4. The legend for Supplementary Figure 1D should be written to specify the genotype of the female flies used in these crosses.

5. The human transgene data from Figure 2B should be plotted with the data in 2C. As presented, it is not obvious that the 29˚C data in 2C show a rescue compared with the 25˚C data in 2B.

6. In Figure 3, the authors use an EP insertion line to show that Dis3L2 overexpression in wing disc compartments reduces compartment size. Do the UAS-Dis3L2 transgenes that were generated for this study also reduce compartment size when expressed under control of en-GAL4 or 69B-GAL4? Also: representative images of wings for B and C should be included, either in the main text or as a supplemental figure.

7. The log scale in Figure S4A obscures the differences in 3’-UTR length between transcripts misexpressed in Dis3L2 mutants compared with controls. Also, is this difference in 3’-UTR length statistically significant?

8. What is the GAL4 driver used for Figure 5B?

9. In Figure S5B, the authors show that re-expression of Dis3L2 in dis3L2 mutant wing discs reduces Idgf2 mRNA. Data for controls – dis3L2 mutants alone – should be added for comparison.

10. Cell types should be indicated in the legend for Supplemental Figure 6, panels A and B.

**Have all data underlying the figures and results presented in the manuscript been provided?**

Reviewer #1: Yes

Reviewer #2: Yes

Reviewer #3: None

PLOS authors have the option to publish the peer review history of their article (what does this mean?). If published, this will include your full peer review and any attached files.

Reviewer #1: No

Reviewer #2: No

Reviewer #3: Yes: Michelle L. Bland

---

## [Decision Letter · Decision Letter 1]

29 Oct 2020

Dear Dr Newbury,

Thank you very much for submitting your Research Article entitled 'Dis3L2 regulates cellular proliferation through a PI3-Kinase dependent signalling pathway' to PLOS Genetics. Your manuscript was fully evaluated at the editorial level and by independent peer reviewers. The reviewers appreciated the attention to an important topic but identified some aspects of the manuscript that should be improved.

We therefore ask you to modify the manuscript according to the review recommendations before we can consider your manuscript for acceptance. Your revisions should address the specific points made by each reviewer.

[LINK]

Yours sincerely,

Ville Hietakangas

Associate Editor

PLOS Genetics

Gregory Barsh

Editor-in-Chief

PLOS Genetics

All reviewers consider that the manuscript has been improved by the revisions. One of the Reviewers still has significant concerns about the rigor of the experiments that support the conclusions about the role of PI3K signaling. One of the concerns is that the quantification of phosphorylation has not been normalized with the total level of the same protein (for example pAKT normalized with AKT). This is especially important, since the changes that you detect are moderate. Since you already have the data on total protein expression (Sup Fig 6C) this should be possible without further experimentation. A second concern is related to the lack of evidence for wing disc specific change in PI3K/AKT signaling. Also another Reviewer comments on this matter. After possible revisions, I recommend you to adjust the strength of your conclusions and bring up the possible remaining caveats. I will assess the revised version, judging whether conclusions of the manuscript are sufficiently supported by experimental evidence, taking into consideration the reviewer feedback and your responses.

Reviewer's Responses to Questions

**Comments to the Authors:**

Reviewer #1: The authors have addressed all my comments and I think the paper is ready for publication

I only had a couple of minor comments:

1. in the methods, the authors described doing western blots on imaginal discs. Did they include these results in the paper? I think the new pAkt data from Fig 7E on whole larvae would be bolstered by complementary blots on discs (if these expts were done).

2. Many blots are heavily cropped with only a thin sliver of each blot with the relevant bands being shown. In these cases, I think its always useful to have images of the uncropped versions as supplemental.

Reviewer #2: The authors have assessed all my concerns and the paper is ready for publication in it current estate.

Reviewer #3: In the revised manuscript, “Dis3L2 regulates cellular proliferation through a PI3-Kinase dependent signaling pathway", Towler et al. provide new data that strongly support their findings of an overgrowth phenotype in Drosophila wing imaginal discs lacking Dis3L2 and convincingly show that the growth factor Idgf2 is a strong driver of growth in Dis3L2 mutant tissue. In particular, very nice, new data show that the overgrowth phenotype of Dis3L2 mutants is largely due to an increase in cell number, with only a 2% increase in cell size.

Statistical tests are lacking for these data:

Sup Figure 1D

Sup Figure 2C – no statistics are shown comparing compartment areas, although the text refers to significant differences or lack of differences (lines 157-158 and 167-170).

Figures 6A, 6B and 6E

The authors’ conclusion that Dis3L2 affects growth via PI3K signaling is not well supported by the data presented, as described below.

First, Western blots shown in Figures 6 and 7 do not include total levels of Akt, PRAS40, MAPK, 4E-BP, etc. for each sample. Second, the phosphorylation levels are normalized by comparing Dis3L2 mutant/knockdown animals/tissue to controls. Instead, levels of phospho-Akt, for example, should be normalized to levels of total Akt in each genotype, and this should be plotted. It is still unclear that apparent changes in phospho-protein levels are driven by activation of the pathway through signaling rather than by changes in total kinase or substrate levels.

Second, wortmannin seems to reduce proliferation of HEK293 cells with Dis3L2 knockdown but has little effect on Akt or PRAS40 phosphorylation. A dose response curve would be helpful in showing that wortmannin effectively inhibits PI3K here.

Third, modest changes in phospho-Akt (Thr342) in whole Dis3L2 mutant larvae are presented as evidence that PI3K signaling is activated in these animals. Total Akt levels are needed to interpret this data, but another problem is that whole-animal changes in Akt may reflect systemic endocrine changes in Dis3L2 mutants. Given the authors’ work with imaginal discs, I proposed that they measure either phospho-Akt or tGPH using immunofluorescence in discs. This could be done in a compartment specific manner, and it would help to address whether changes in Akt phosphorylation might occur tissue- or cell-autonomously in response to Dis3L2 or Idgf2 manipulations.

At present, other possibilities, including parallel, independent growth pathways regulated by PI3K and Dis3L2, could explain the results presented here and therefore the title should be reconsidered.

**Have all data underlying the figures and results presented in the manuscript been provided?**

Reviewer #1: Yes

Reviewer #2: Yes

Reviewer #3: Yes

PLOS authors have the option to publish the peer review history of their article (what does this mean?). If published, this will include your full peer review and any attached files.

Reviewer #1: No

Reviewer #2: No

Reviewer #3: No

---

## [Editor Report · Decision Letter 2]

5 Dec 2020

Dear Dr Newbury,

We are pleased to inform you that your manuscript entitled "Dis3L2 regulates cell proliferation and tissue growth though a conserved mechanism" has been editorially accepted for publication in PLOS Genetics. Congratulations!

Yours sincerely,

Ville Hietakangas

Associate Editor

PLOS Genetics

Gregory Barsh

Editor-in-Chief

PLOS Genetics

Comments from the reviewers (if applicable):

**Data Deposition**

http://datadryad.org/submit?journalID=pgenetics&manu=PGENETICS-D-20-00279R2

**Press Queries**

---

## [Editor Report · Acceptance letter]

23 Dec 2020

PGENETICS-D-20-00279R2 

Dis3L2 regulates cell proliferation and tissue growth though a conserved mechanism 

Dear Dr Newbury, 

We are pleased to inform you that your manuscript entitled "Dis3L2 regulates cell proliferation and tissue growth though a conserved mechanism" has been formally accepted for publication in PLOS Genetics! Your manuscript is now with our production department and you will be notified of the publication date in due course.

With kind regards,

Melanie Wincott

PLOS Genetics

On behalf of:
